# EQUIVARIANT VECTOR FIELD NETWORK FOR MANY-BODY SYSTEM MODELING

## ABSTRACT

Modeling many-body systems has been a long-standing challenge in science, from classical and quantum physics to computational biology. Equivariance is a critical physical symmetry for many-body dynamic systems, which enables robust and accurate prediction under arbitrary reference transformations. In light of this, great efforts have been put on encoding this symmetry into deep neural networks, which significantly boosts the prediction performance of down-streaming tasks. Some general equivariant models which are computationally efficient have been proposed, however, these models have no guarantee on the approximation power and may have information loss. In this paper, we leverage insights from the scalarization technique in differential geometry to model many-body systems by learning the gradient vector fields, which are SE(3) and permutation equivariant. Specifically, we propose the Equivariant Vector Field Network (EVFN), which is built on a novel tuple of equivariant basis and the associated scalarization and vectorization layers. Since our tuple equivariant basis forms a complete basis, learning the dynamics with our EVFN has no information loss and no tensor operations are involved before the final vectorization, which reduces the complex optimization on tensors to a minimum. We evaluate our method on predicting trajectories of simulated Newton mechanics systems with both full and partially observed data, as well as the equilibrium state of small molecules (molecular conformation) evolving as a statistical mechanics system. Experimental results across multiple tasks demonstrate that our model achieves best or competitive performance on baseline models in various types of datasets.

## 1 INTRODUCTION

Modeling many-body systems has been a long-standing challenge in scientific fields from classical and quantum physics (Carleo & Troyer, 2017; Zhang et al., 2018; Satorras et al., 2021c) to structural biology (Senior et al., 2020; Shi et al., 2021), due to its high numerical complexity and complicated evolving mechanism. Graph neural network (GNN), which is superior to model the high-dimensional structured data with permutation equivariance, brings a new opportunity to model the many-body systems in an end-to-end manner. Since many-body physical systems follow many physical constraints like SE(3) symmetry, pure black-box GNN models show limitations on generalization in this scenario and symmetry-preserving GNN models have become a hot research direction.

The core question to be solved for developing general equivariant GNN models is how to conduct nonlinear operations on tensors in a reference-free way. To represent and manipulate equivariant tensor of arbitrary orders, some approaches resort to equivariant function spaces such as spherical harmonics (Thomas et al., 2018; Fuchs et al., 2020; Bogatskiy et al., 2020; Fuchs et al., 2021) or lifting the spatial space to high-dimensional spaces such as Lie group space (Cohen & Welling, 2016; Cohen et al., 2018; 2019b; Finzi et al., 2020; Hutchinson et al., 2021). Since no restriction on the order of tensors is imposed on these methods, sufficient expressive power of these models is guaranteed. Unfortunately, transforming a many-body system into those high-dimensional spaces or calculating equivariant functions usually brings excessive computational cost and great optimization difficulty, which is unacceptable in some real-world scenarios. To remedy this issue, Satorras et al. (2021c) proposed EGNN to directly implement equivariant operations in the original space, providing an efficient way to preserve equivariance without performing complex space transformations.

Detailed experiments in (Satorras et al., 2021c) have shown that preserving equivariance without transforming the space is theoretically possible and computationally efficient in practice.

However, one trade-off of EGNN is abandoning a certain amount of tensor information [1], which causes the equivariance function class EGNN can approximate to be restricted. This drawback may become serious when modeling complex dynamical scenarios such as molecular simulation, where geometric information (e.g., angular potentials and rotatable bonds) plays an important role in inducing conformation changes (Klicpera et al. (2020); Xu et al. (2021)). To mitigate this issue, we propose a new model called Equivariant Vector Field Network (EVFN) to fit the gradient vector fields (Song & Ermon, 2019; Shi et al., 2021) of many-body systems. With a scalarization block and a vectorization block, EVFN is able to represent tensor information losslessly in the original space and outputs equivariant vector fields with no restriction on the direction.

Inspired by the scalarization technique from differential geometry (Kobayashi, 1963; Hsu, 2002), EVFN first introduces a tuple of complete basis associated with each particle pair that preserves permutation and SE(3) symmetry under global rotation and translation transformation on the many-body system. Based on this basis, the scalarization block losslessly transforms the geometric information into SO(3)-invariant scalar representations. In principle, the scalar representations can be fed into any permutation-equivariant network to implement complex nonlinear transformations. Moreover, the vectorization block could reverse the scalars back to the vector field without sacrificing geometric information with the complete basis. Once obtained the estimated gradient field, we can predict a certain state or the whole dynamical trajectory of a 3D many-body system via an integration procedure.

We evaluate the proposed framework on two many-body scenarios that require equivariance: (1) the simulated Newtonian many-body dynamics trajectory prediction and (2) the real-world molecular conformation generation. Our model achieves best or competitive results in various types of datasets.

## 2 BACKGROUND

In this section, we first introduce some basic concepts on the notion of equivariance and tensor field and then describe the scalarization technique from differential geometry. Finally, we define the 'vector field' as the differential of a many-body system. Let $\boldsymbol{X} = (\boldsymbol{x}_1, \ldots, \boldsymbol{x}_N) \in \mathbb{R}^{N \times 3}$ be a many-body system living in $\mathbb{R}^3$, where $N$ is the number of particles. We use $\boldsymbol{x}_i(t)$ to denote the position of the particle $i$ at time $t$.

**SE(3) group and Equivariance** In the Euclidean space $\mathbb{R}^3$ we can consider affine transformations that preserve the distance between any two points, i.e., the isometric group SE(3). We call it the symmetry group w.r.t. the Euclidean metric, and it turns out that SE(3) can be generated by the translation group and the rotation group SO(3).

Once we have a symmetry group, it's valid to define quantities that are "equivariant" under the symmetry group. Given a function $f : \mathbb{R}^m \to \mathbb{R}^n$, assuming the symmetry group $G$ acts on $\mathbb{R}^m$ and $\mathbb{R}^n$, then $f$ is $G$-**equivariant** if

$$f(gx) = gf(x), \quad \forall x \in \mathbb{R}^m \text{ and } g \in G.$$

For SO(3) group, if $n = 1$, i.e., the output of $f$ is a scalar, then the group action on $\mathbb{R}^1$ is the identity map, in this case $f$ should be $SO(3)$-invariant (Thomas et al., 2018): $f(gx) = f(x)$.

The notion of **tensor field** can be defined for general Riemannian manifold (see Definition 2.1.10 of (Jost & Jost, 2008)). Let $\{\frac{\partial}{\partial x_i}\}_{i=1}^3$ and $\{dx^i\}_{i=1}^3$ be the tangent vectors and dual vectors in $\mathbb{R}^3$ respectively, and $\otimes$ denotes the tensor product. Then recall the definition of tensor field for $\mathbb{R}^3$ w.r.t. the SO(3) group:

**Definition 2.1.** *A (r, s)- tensor field $\theta$ is a multi-linear map from a collection of r vectors and s dual vectors in $\mathbb{R}^3$ to $\mathbb{R}$: $\theta(x) = \theta_{j_1 \cdots j_s}^{i_1 \cdots i_r} \frac{\partial}{\partial x_{i_1}} \otimes \cdots \otimes \frac{\partial}{\partial x_{i_r}} \otimes dx^{j_1} \otimes \cdots \otimes dx^{j_s}$. It implies that under SO(3) coordinate transformation $g := \{g_{ij}\}_{1 \leq i,j \leq n}$, the tensor field $\theta$ transforms equivariantly: $\theta_{j_1' \cdots j_s'}^{i_1' \cdots i_r'} = g_{i_1' i_1} \cdots g_{i_r' i_r} g_{j_1 j_1'}^T \cdots g_{j_s j_s'}^T \theta_{j_1 \cdots j_s}^{i_1 \cdots i_r}$, where $g^T$ is the inverse of g.*

---

[1] For example, dihedral angle is a function of position vectors rather than the position's norm, which are the input of EGNN. Detailed definition is given in appendix A.2.4

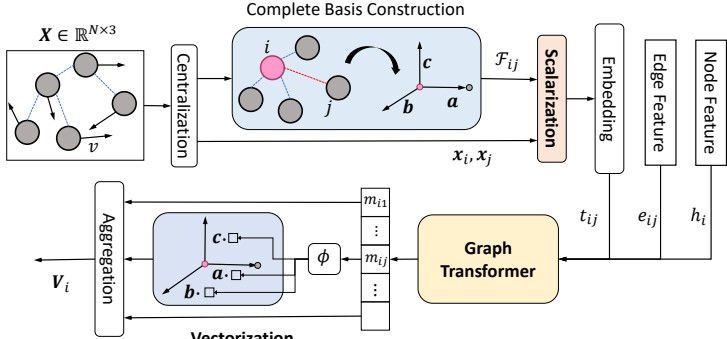

Figure 1: An overview of EVFN. For a many-body system $\boldsymbol{X}$, we first centralize the positions to preserve translation equivariance. Then we introduce a tuple of edge-level complete basis $\mathcal{F}_{ij}$ to transform the geometric tensors $\boldsymbol{x}_i$ into SO(3)-invariant scalars. Afterwards, the scalar embeddings $t_{ij}$, pre-defined node features $h_i$ and edge features $e_{ij}$ are fed to the Graph Transformer to learn edgewise embeddings $m_{ij}$. Finally, a vectorization block transforms the edgewise embeddings into nodewise vector fields $\mathbf{V}_i$.

**Equivariant Vector field**    To model $\boldsymbol{X}(t)$, a natural way is to estimate its differential $\frac{d\boldsymbol{X}(t)}{dt}$ and apply an ODE solver to integrate the differential to obtain the dynamic trajectory or a state at a given time. Due to the SO(3) symmetry, we define such differential as an **equivariant vector field**. Most 3D real-world scenarios adopt **first-order** and **second-order** equivariant vector fields to depict their dynamic evolving mechanisms, which are also the modeling targets in this paper. A typical second-order vector field is the acceleration field of Newtonian systems.

**Gradient field** is a widely used terminology meaning the first-order derivative w.r.t. a scalar function (Jost & Jost, 2008; Song & Ermon, 2019). To generate molecular conformations (i.e. equilibrium state) with a single stage, (Shi et al., 2021) define "gradient field" to serve as pseudo-forces acting on each particle. By evolving the particles following the direction of the gradient field, the non-equilibrium system will finally converge to an equilibrium state. Gradient field is a special case of **first-order** vector field.

## 3  METHODOLOGY

Given a many-body system $\boldsymbol{X}$, we aim at modeling its vector field to predict the long-term dynamic trajectory or the equilibrium state within a single stage. To preserve the physical symmetries of the system, the estimated vector field should be equivariant for permutation and SE(3) group. To achieve this goal, we represent the system as a spatial graph and construct **EVFN** based on it with three key components: (1) a **Scalarization** block to encode the geometric tensors into SO(3)-invariant scalar representations attached to each node; (2) a **Graph Transformer** block to learn SO(3)-invariant edgewise embeddings by propagating and aggregating information on the graph; and (3) a **Vectorization** block to reverse scalar representations back to geometric tensors to estimate the vector field. A brief overview of EVFN is illustrated in Figure 1. Once the vector field network (EVFN) is optimized, an **Evolving** block is incorporated to integrate the vector field for predicting the dynamics. The translation symmetry can be easily preserved by moving the particle system's centroid at $t = 0$ to the origin (the **Centralization** operation in Figure 1). Permutation equivariance is automatically guaranteed for the message-passing based Graph Transformer. We provide detailed proof about these symmetries in Appendix A.2.1. Now we concentrate on SO(3) symmetry in the following sections.

### 3.1  SCALARIZATION BLOCK

The scalarization block is designed to transform geometric tensors into edgewise SO(3)-invariant scalar features by introducing a novel tuple of complete basis. Given a particle $\boldsymbol{x}_i(t)$, define neighbor

$\mathcal{N}(\boldsymbol{x}_i(t))$ as the particles that react with $\boldsymbol{x}_i(t)$. Then we can consider a particle pair $(\boldsymbol{x}_i(t), \boldsymbol{x}_j(t))$, where $\boldsymbol{x}_j(t) \in \mathcal{N}(\boldsymbol{x}_i(t))$. Suppose we take the positions of the two particles as the relevant geometric information of edge $\langle i, j \rangle$, then the edgewise SO(3)-invariant scalars could be defined as $t_{ij} := \text{Scalarize}(\boldsymbol{x}_i(t), \boldsymbol{x}_j(t), \mathcal{F}_{ij})$, where Scalarize is the scalarization operation under an edgewise dynamical basis $\mathcal{F}_{ij}$ defined below.

**Equivariant basis construction**  For the particle pair $(\boldsymbol{x}_i(t), \boldsymbol{x}_j(t))$, let $\boldsymbol{a}(t) = \frac{\boldsymbol{x}_i(t) - \boldsymbol{x}_j(t)}{\|\boldsymbol{x}_i(t) - \boldsymbol{x}_j(t)\|}$ and $\times$ denotes the cross product of two vectors, define

$$\boldsymbol{b}(t) = \frac{\boldsymbol{x}_i(t) \times \boldsymbol{x}_j(t)}{\|\boldsymbol{x}_i(t) \times \boldsymbol{x}_j(t)\|} \quad \text{and} \quad \boldsymbol{c}(t) = \boldsymbol{a}(t) \times \boldsymbol{b}(t), \tag{3.1}$$

Then we build a SO(3)-equivariant basis $\mathcal{F}_{ij} := (\boldsymbol{a}(t), \boldsymbol{b}(t), \boldsymbol{c}(t))$. In practice we add a small constant $\epsilon$ to the normalization factor in case that $\boldsymbol{x}_i$ and $\boldsymbol{x}_j$ collapse. Under the condition that the matrix $(\boldsymbol{a}(t), \boldsymbol{b}(t), \boldsymbol{c}(t))$ is non-degenerate, $\mathcal{F}_{ij}$ formulates a complete orthonormal basis (frame) of the tangent space at $\boldsymbol{x}_i(t)$. Note that this is a dynamical basis w.r.t. $t$ and the construction process of such basis is permutation-equivariant. Since the Euclidean metric is flat, the dual basis (living in the cotangent space of $\boldsymbol{x}_i$ (Hsu, 2002)) of $\mathcal{F}_{ij}$ is just its transpose: $(\boldsymbol{a}^T(t), \boldsymbol{b}^T(t), \boldsymbol{c}^T(t))$. The 'bad' event that $\mathcal{F}_{ij}$ is degenerate for all neighbors happens only when all particles are restricted to a straight line, which is a measure zero set in $\mathbb{R}^3$. Therefore, we assume $\mathcal{F}_{ij}$ is non-degenerate from now on. Proof for SO(3)-equivariance of $\mathcal{F}_{ij}$ is provided in proposition A.1.

**Invariant scalarization of geometric tensors**  With the complete equivariant basis, we can realize the scalarization operation (Kobayashi, 1963) in an elementary way. First of all, notice that under the basis $\mathcal{F}_{ij}$ $((\boldsymbol{a}(t), \boldsymbol{b}(t), \boldsymbol{c}(t)))$, the position vector of $\boldsymbol{x}_k$ naturally owns a 'coefficient' or 'scalar' representation:

$$(\boldsymbol{x}_k \cdot \boldsymbol{a}(t), \boldsymbol{x}_k \cdot \boldsymbol{b}(t), \boldsymbol{x}_k \cdot \boldsymbol{c}(t)). \tag{3.2}$$

We define the process obtaining such scalars as Scalarize operation. Here we demonstrate that the set of obtained coefficients (3.2) is actually a SO(3)-invariant scalar tuple. Let $g \in SO(3)$ be an arbitrary orthogonal transformation, then

$$\boldsymbol{x}_k \to g\boldsymbol{x}_k \quad \text{and} \quad (\boldsymbol{a}(t), \boldsymbol{b}(t), \boldsymbol{c}(t)) \to (g \cdot \boldsymbol{a}(t), g \cdot \boldsymbol{b}(t), g \cdot \boldsymbol{c}(t)).$$

Therefore (3.2) undergoes

$$(\boldsymbol{x}_k \cdot \boldsymbol{a}(t), \boldsymbol{x}_k \cdot \boldsymbol{b}(t), \boldsymbol{x}_k \cdot \boldsymbol{c}(t)) \to (g\boldsymbol{x}_k \cdot g\boldsymbol{a}(t), g\boldsymbol{x}_k \cdot g\boldsymbol{b}(t), g\boldsymbol{x}_k \cdot g\boldsymbol{c}(t))$$
$$= (\boldsymbol{x}_k \cdot \boldsymbol{a}(t), \boldsymbol{x}_k \cdot \boldsymbol{b}(t), \boldsymbol{x}_k \cdot \boldsymbol{c}(t)), \tag{3.3}$$

where we use the fact that $g^T g = I$ to get the last line.

It is easy to prove that the Scalarize operation could transform arbitrary geometric tensors into SO(3)-invariant scalars. Taking $(2,0)$-type tensors as an example, by extending the complete basis $(\boldsymbol{a}, \boldsymbol{b}, \boldsymbol{c})$ through tensor product, it's easy to check that

$$\{\boldsymbol{a} \otimes \boldsymbol{a}, \boldsymbol{b} \otimes \boldsymbol{b}, \boldsymbol{c} \otimes \boldsymbol{c}, \boldsymbol{a} \otimes \boldsymbol{b}, \boldsymbol{b} \otimes \boldsymbol{a}, \boldsymbol{a} \otimes \boldsymbol{c}, \boldsymbol{c} \otimes \boldsymbol{a}, \boldsymbol{b} \otimes \boldsymbol{c}, \boldsymbol{c} \otimes \boldsymbol{b}\}$$

forms an equivariant basis of the $(2,0)$-type tensor space. Then the scalarization of a tensor is just the linear combination coefficients under this basis. In the same way as (3.3), we can prove that the coefficients are also SO(3)-invariant scalars. Given a $(2,0)$-symmetric tensor $\theta$ (e.g., energy-momentum tensor), under the complete basis $\mathcal{F}_{ij} = (\boldsymbol{a}, \boldsymbol{b}, \boldsymbol{c})$, $\theta$ can be expressed as:

$$\theta = \theta^{aa} \boldsymbol{a} \otimes \boldsymbol{a} + \theta^{bb} \boldsymbol{b} \otimes \boldsymbol{b} + \theta^{cc} \boldsymbol{c} \otimes \boldsymbol{c} + \theta^{ab}(\boldsymbol{a} \otimes \boldsymbol{b} + \boldsymbol{b} \otimes \boldsymbol{a})$$
$$+ \theta^{ac}(\boldsymbol{a} \otimes \boldsymbol{c} + \boldsymbol{c} \otimes \boldsymbol{a}) + \theta^{bc}(\boldsymbol{b} \otimes \boldsymbol{c} + \boldsymbol{c} \otimes \boldsymbol{b}). \tag{3.4}$$

The scalars tuple $t_{ij} := \{\theta^{aa}, \theta^{ab}, \dots\}$ are the scalarization of $\theta$ under the equivariant basis $\mathcal{F}_{ij}$, which are SO(3)-invariant. Since any nonlinear transformations acting on SO(3)-invariant scalars are still SO(3)-invariant, therefore the scalars tuple can be fed into **any neural network architectures without concerns about breaking the equivariance symmetry**. Our scalarization is inspired by the scalarization technique on the frame bundle in differential geometry (Hsu, 2002), the equivalence of the scalarization technique and (3.4) is given in proposition A.2.

In practice, we focus on the scalarization of $(1,0)$-type tensors (i.e., vectors). Since the most common geometric information of input is vectors in real-world scenarios. We define the Scalarize operation as:

$$\text{Scalarize}(\boldsymbol{x}_i, \boldsymbol{x}_j, \mathcal{F}_{ij}) = (\boldsymbol{x}_i \cdot \boldsymbol{a}(t), \boldsymbol{x}_i \cdot \boldsymbol{b}(t), \boldsymbol{x}_i \cdot \boldsymbol{c}(t), \boldsymbol{x}_j \cdot \boldsymbol{a}(t), \boldsymbol{x}_j \cdot \boldsymbol{b}(t), \boldsymbol{x}_j \cdot \boldsymbol{c}(t)) \tag{3.5}$$

## 3.2 GRAPH TRANSFORMER BLOCK

After encoding the geometric tensors into SO(3)-invariant scalars $t_{ij}$, we first embed them alone with other pre-defined node/edge attributes ($h_j$, $e_{ij}$) into high-dimensional representations, and leverage an attention-based Graph Transformer architecture (Shi et al., 2020) to learn the SO(3)-invariant edgewise message embeddings $m_{ij}$ by propagating and aggregating information on the graph $\mathcal{G}_X$. The attention mechanism is introduced due to its powerful capacity in modeling those graphs with unknown topology. We provide further design details in Appendix A.2.2.

## 3.3 VECTORIZATION BLOCK

Given the refined edgewise message $m_{ij}$, the vectorization block is designed to transform these scalars back to equivariant vectors, which requires pairing $m_{ij}$ with the corresponding complete basis[2] $\mathcal{F}_{ij} := (\boldsymbol{a}, \boldsymbol{b}, \boldsymbol{c})$. More precisely, we first project $m_{ij}$ into a scalar triple $\{x^1, x^2, x^3\}$, then define the vectorization process as:

$$(x^1, x^2, x^3) \xrightarrow{\text{Pairing}} x^1\boldsymbol{a} + x^2\boldsymbol{b} + x^3\boldsymbol{c}. \qquad (3.6)$$

---

**Algorithm 1: EVFN**

**Input:** $\mathbf{X} = (\boldsymbol{x}_1, \ldots, \boldsymbol{x}_N) \in \mathbb{R}^{N \times 3}$,
$\quad\quad h_i \in \mathbb{R}^h, e_{ij} \in \mathbb{R}^e, \mathcal{G}_X$
`// Centralization`
$\mathbf{X} \leftarrow \texttt{Centralize}(\mathbf{X})$;
`// Scalarization Block`
**for** $\boldsymbol{x}_i \in X$ **do**
$\quad$ **for** $\boldsymbol{x}_j \in \mathcal{N}(\boldsymbol{x}_i)$ **do**
$\quad\quad$ $\mathcal{F}_{ij} = \texttt{EquiBasis}(\boldsymbol{x}_j, \boldsymbol{x}_j)$;
$\quad\quad$ $t_{ij} = \texttt{Scalarize}(\mathcal{F}_{ij}, \boldsymbol{x}_i, \boldsymbol{x}_j)$;
$\quad$ **end**
**end**
`// Graph Transformer Block`
$m_{ij} = \texttt{GraphTransformer}(t_{ij}, h_i, e_{ij})$;
`// Vectorization Block`
$V_i = \frac{1}{N} \sum_{j \in \mathcal{N}(x_i)} \texttt{Vectorize}(m_{ij}, \mathcal{F}_{ij})$;
**Output:** $\mathbf{V}_i$

---

We encapsulate the pairing process as $V_{ij} = \text{Vectorize}(m_{ij}, \mathcal{F}_{ij})$. It's easy to check that the output follows the transformation rule of vectors. Finally, we aggregate all vectors $V_{ij}$ associated with $\boldsymbol{x}_i$ to estimate the ground-truth vector field $V_i$.

So far, we have achieved permutation and SE(3) equivariance by employing the three blocks. The evolving block is introduced to simulate dynamics of the system with the optimized vector field network, acting like an ODE solver. We will discuss the evolving block designed specifically for each scenario in Section 4. The workflow of our method is summarized in Algorithm 1.

**Discussion:** First, the SO(3)-equivariance of EVFN can be proved from the following facts: (1) The scalarization block is SO(3)-invariant because its inputs are SO(3)-invariant scalars; (2) The SO(3)-invariant variables are still SO(3)-invariant after being transformed by the graph transformer block (or any nonlinear neural network); (3) The vectorization block is SO(3)-equivariant if the input is SO(3)-invariant. We put the detailed proof in appendix A.2.3. Second, note that by pairing the scalars obtained from the scalarization block with the equivariant basis, we can recover back the tensor fields in the data. Therefore the transformation is invertible, which implies that there is no information loss when going through the scalarization block.

## 4 EXPERIMENTS

Our method (EVFN together with the evolving block) is a general framework for modeling many-body systems. To validate the effectiveness of the method, we conduct extensive experiments on two scenarios: (1) simulated Newtonian many-body systems trajectory prediction (second-order vector field task) and (2) the real-world molecular conformation prediction (first-order vector field task).

### 4.1 NEWTONIAN MANY-BODY SYSTEM

In this experiment, we apply our model to predict the long-term motion trajectory of an unknown many-body Newton system given its initial position and velocity. To highlight the strength of EVFN on including complete geometric information, we place particular emphasis on non-radical Newtonian forces under three settings.

---

[2]Since every vector is a linear combination of a basis, the output of the vectorization block has no restriction on the direction. We will discuss how our basis changes under reflection in remark A.3.

**Partially observed system (POS).** This system consists of six particles under Newton's gravitation (radical force) but only four of them could be observed, i.e., for each trajectory, we are provided with positions $\boldsymbol{X}(t) \in \mathbb{R}^{4 \times 3}$ and velocities $\boldsymbol{V}(t) \in \mathbb{R}^{4 \times 3}$. The two unobserved particles act as a 'virtual' external field for the system.

**Static external force field (SEFF).** This system consists of six particles under both Newton's gravitation and an external static force : $\boldsymbol{f}_\eta = (0, 0, \eta)$, where $\eta > 0$ is the magnitude of the external field along the z-axis.

**Dynamical external force field (DEFF).** This system consists of three particles governed by both Newton's gravitation and a Lorentz-like dynamical external force field, which means there exists a force field perpendicular to the current direction of each particle's velocity: $\boldsymbol{f}_l(\boldsymbol{v}(t)) = q\boldsymbol{v}(t) \times \mathbf{B}$, where $q$ is a positive constant that mimics the charge of particles and $\mathbf{B}$ denotes the pseudo-vector of the external field.

**Problem definition.** Following (Zhuang et al., 2020; Li et al., 2021), we formulate trajectory prediction as two tasks: **Interpolation** and **Extrapolation** in the original and rotated reference.

The experimental setting is as follows. For each trajectory, given the initial condition, we use observations $\boldsymbol{x}_i(t), t \in \{\Delta t, 2\Delta t \ldots, T_1\}$ as the training labels and observations $\boldsymbol{x}_i(t), t \in \{T_1 + \Delta t, T_1 + \Delta t, \ldots, T_2\}$ as the validation set. To evaluate the interpolation and extrapolation capacity of all methods, the observations $\boldsymbol{x}_i(t), t \in \{\frac{1}{2}\Delta t, \frac{3}{2}\Delta t, \ldots, T_1 + \frac{1}{2}\Delta t\}$ and $\boldsymbol{x}_i(t), t \in \{T_2 + \Delta t, T_2 + 2\Delta t, \ldots, T_3\}$ are used as **interpolation** and **extrapolation** test sets respectively. We measure the mean square error (MSE) between the predicted trajectory and ground truth for both tasks. To measure the exactness of equivariance, we follow (Fuchs et al., 2020) to apply uniformly sampled SO(3)-transformations on the input and output. The MSE between the predicted trajectory with rotated input and rotated ground truth could reflect the transformation robustness of method. The normalized distance between the rotated prediction with original input and the original prediction with the rotated input defines the equivariance error $\Delta_{EQ}$:

$$\Delta_{EQ} = \|L_s\Phi(\boldsymbol{x}) - \Phi L_s(\boldsymbol{x})\|/\|L_s\Phi(\boldsymbol{x})\|, \tag{4.1}$$

where $L_s$ and $\Phi$ denote SO(3) transformations and equivariant neural networks, respectively.

**Learning Framework.** Inspired by (Norcliffe et al., 2020), for a Newtonian system, we utilize EVFN to parameterize its acceleration vector field and adopt a second-order neural ODE (SNODE) as the evolving block (see Appendix A.2.5) to integrate both the position and velocity trajectories. Only the MSE between predicted position trajectory and ground truth is taken as the loss penalty:

$$\mathcal{L}(\theta) = \frac{1}{n}\sum_{i=1}^{n} L_2(x_{t_i}, \text{SNODE}(\boldsymbol{x}_{t_0}, \boldsymbol{v}_{t_0}, t_0, t_i, \Theta)), \ \ t_0 < t_1 < \cdots < t_n, \tag{4.2}$$

where $(\boldsymbol{x}_{t_0}, \boldsymbol{v}_{t_0})$ and $\Theta$ denote the initial condition of the system and the parameters of EVFN $\Phi$.

**Implementation Details.** Following (Zhuang et al., 2020), all trajectories are simulated using the *Dopri5* solver (Dormand & Prince, 1980) with the tolerance to $10^{-7}$ and the modified physical rules. In this experiment, we set $\eta$ to 0.98, $q$ to 1 and $\mathbf{B}$ to $[0.5, 0.5, 0.5]^\top$, respectively. We sample 1, 100 and 100 trajectories for the three systems as our evaluation platform, where the data is split into training, validation and test sets by the time span: $T_1 = \{1.5, 0.5, 0.5\}$, $T_2 = \{2, 0.55, 0.55\}$ and $T_3 = \{2.5, 0.6, 0.6\}$. We compare our method to the non-equivariant graph convolutional network (GCN) (Kipf & Welling, 2016) without and with SO(3) data augmentation (denoted as GCN (Aug)), as well as two equivariant models designed for vector field modeling: Radial Field (Köhler et al., 2019) and EGNN (Satorras et al., 2021a;c). Further details about data generation and model implementation are provided in Appendix A.3.1.

**Results.** All the baselines and EVFN implement the feature transformations in the original space. Notice that GNNs are manifestly permutation-equivariant and the centroid of data is reduced for all models to preserve translation equivariance. Thus we only need to evaluate the generalization capacity of all models for SO(3) transformations.

From Table 1, we have the following conclusions: (1) With the original input, EVFN outperforms all other equivariant methods in the interpolation and extrapolation tasks, which demonstrates that EVFN exhibits stronger expressive power by representing geometric information losslessly[3];

---

[3]We put more discussions especially the performance of the non-equivariant GCN in Appendix A.3.1.

Table 1: Comparison on three simulated Newtonian systems. *Inter.* and *Extra.* denote the interpolation and extrapolation task respectively. *Rot.* denotes the results in the rotated reference.

| Setting | Method | Inter. | Extra. | Rot. Inter. | Rot. Extra. | $\Delta_{EQ}$ |
|---------|--------|--------|--------|-------------|-------------|---------------|
| POS | GCN | 0.143 | **1.452** | 0.772 | 2.633 | 18.163 |
| | GCN (Aug) | 1.177 | 6.696 | 1.821 | 14.310 | 18.163 |
| | Radial Field | 1.996 | 7.665 | 1.996 | 7.665 | $5.77 \cdot 10^{-6}$ |
| | EGNN | 0.726 | 6.449 | 0.726 | 6.462 | $\mathbf{9.19 \cdot 10^{-7}}$ |
| | EVFN | **0.138** | 2.502 | **0.137** | **2.428** | $1.23 \cdot 10^{-4}$ |
| | EVFN* | 0.206 | 2.432 | 0.206 | 2.442 | $4.13 \cdot 10^{-6}$ |
| SEFF | GCN | $\mathbf{1.24 \cdot 10^{-2}}$ | **0.119** | 0.462 | 3.824 | 0.154 |
| | GCN (Aug) | 0.173 | 1.389 | 0.463 | 3.931 | 0.154 |
| | Radial Field | 0.701 | 6.032 | 0.701 | 6.032 | $\mathbf{5.23 \cdot 10^{-7}}$ |
| | EGNN | 0.463 | 4.248 | 0.467 | 4.275 | $5.88 \cdot 10^{-6}$ |
| | EVFN | 0.139 | 1.289 | **0.142** | **1.310** | $1.05 \cdot 10^{-6}$ |
| | EVFN* | 0.278 | 2.406 | 0.289 | 2.556 | $8.17 \cdot 10^{-7}$ |
| DEFF | GCN | $2.99 \cdot 10^{-3}$ | $4.75 \cdot 10^{-2}$ | 0.203 | 1.524 | $8.86 \cdot 10^{-2}$ |
| | GCN (Aug) | $8.81 \cdot 10^{-2}$ | 0.740 | $9.07 \cdot 10^{-2}$ | 0.760 | $8.86 \cdot 10^{-2}$ |
| | Radial Field | $9.57 \cdot 10^{-2}$ | 0.804 | $9.57 \cdot 10^{-2}$ | 0.804 | $\mathbf{5.10 \cdot 10^{-7}}$ |
| | EGNN | $1.99 \cdot 10^{-2}$ | 0.213 | $3.22 \cdot 10^{-2}$ | 0.290 | $8.55 \cdot 10^{-7}$ |
| | EVFN | $\mathbf{1.38 \cdot 10^{-3}}$ | $\mathbf{2.43 \cdot 10^{-2}}$ | $\mathbf{1.39 \cdot 10^{-3}}$ | $\mathbf{2.48 \cdot 10^{-2}}$ | $1.15 \cdot 10^{-5}$ |
| | EVFN* | $1.66 \cdot 10^{-3}$ | $3.13 \cdot 10^{-2}$ | $1.81 \cdot 10^{-3}$ | $3.24 \cdot 10^{-2}$ | $6.51 \cdot 10^{-7}$ |

(2) With the rotated input, EVFN still performs best and all equivariant networks exhibit better equivariance-preserving capacity than GCNs even with data augmentation, implying that it is difficult to achieve equivariance with simple data augmentation. We leave a detailed analysis about different augmentation degrees in Appendix A.3.1. [4]

## 4.2 MOLECULAR CONFORMATION GENERATION

In this experiment, we leverage EVFN to fit the first-order vector field of small molecules' equilibrium distribution and generate reasonable conformations for a given molecular graph. Note that in this case, the SO(3) and translation symmetry are in the distribution level, rather than a specific trajectory. Previous works demonstrate that geometric information (e.g., angular potentials and rotatable bonds) plays a critical role in inducing conformation changes (Xu et al., 2021; Klicpera et al., 2020), which provides a natural platform to evaluate the strength of EVFN.

**Evaluation Tasks.** We conduct experiments on two tasks: (1) **Conformation Generation** evaluates the capacity of EVFN to learn the conformation distribution by measuring the diversity and accuracy of generated conformations. (2) **Distributions Over Distances** evaluates the discrepancy of distance geometry between the generated conformations and the reference conformations.

**Learning Framework.** Following (Shi et al., 2021), for this first-order statistical ensemble system, we leverage a score-based generative modeling framework to estimate the gradient field of atomic positions (See more details about score-based networks in (Shi et al., 2021; Song et al., 2020) or Appendix A.2.5, A.3.2). The optimization objective of EVFN $\Phi$ can be summarized as:

$$\mathcal{L}(\theta) = \frac{1}{n} \sum_{i=1}^{n} \mathbb{E}_{\boldsymbol{X}(0)} \{\lambda(t_i) \| \nabla H_{t_i}(\boldsymbol{X}(t_i)) - \Phi(\boldsymbol{X}(t_i), t_i, \Theta) \|_2^2, \ t_0 < t_1 < \cdots < t_n, \quad (4.3)$$

where $\lambda(t) : [0, T] \to \mathbb{R}^+$ is a positive weighting function and $\nabla H_{t_i}$ is the pre-computed gradient field of noisy atomic positions. Once the score network is optimized, we can use an annealed Langevin dynamics (ALD) sampler or an ODE-based PC sampler as the evolving block to generate conformations (Song et al., 2020).

---

[4]Note that the division operation in the basis construction procedure will bring considerable numerical error into EVFN which increases $\Delta_{EQ}$. Therefore, we tune the constant $C$ in $\boldsymbol{a}(t) = \frac{\boldsymbol{x}_i(t) - \boldsymbol{x}_j(t)}{\|\boldsymbol{x}_i(t) - \boldsymbol{x}_j(t)\| + C}$. The model variant EVFN* with the suitable $C$ can achieve a comparable equivariant error with other baselines.

Table 2: COV and MAT scores of different approaches on GEOM-QM9 and GEOM-Drugs datasets. For the COV score, the threshold $\delta$ is set to $0.5$Å for QM9 and $1.25$Å for Drugs.

| Dataset | GEOM-QM9 | | | | GEOM-Drugs | | | |
|---|---|---|---|---|---|---|---|---|
| Metric | COV (%)↑ | | MAT (Å)↓ | | COV (%)↑ | | MAT (Å)↓ | |
| | Mean | Median | Mean | Median | Mean | Median | Mean | Median |
| RDKit | 83.26 | 90.78 | 0.3447 | 0.2935 | 60.91 | 65.70 | 1.2026 | 1.1252 |
| CGCF | 78.05 | 82.48 | 0.4219 | 0.3900 | 53.96 | 57.06 | 1.2487 | 1.2247 |
| ConfGF | 88.49 | **94.13** | 0.2673 | 0.2685 | 62.15 | 70.93 | 1.1629 | 1.1596 |
| EGNN | 80.93 | 86.27 | 0.3832 | 0.3898 | 40.71 | 33.01 | 1.3574 | 1.3346 |
| EVFN | **90.21** | 93.14 | **0.2430** | **0.2457** | **88.64** | **97.56** | **0.9040** | **0.9023** |

**Datasets.** Following (Xu et al., 2021; Shi et al., 2021) we evaluate the proposed model on the GEOM-QM9 and GEOM-Drugs datasets (Axelrod & Gomez-Bombarelli, 2020) as well as the ISO17 dataset (Schütt et al., 2017). To keep a fair comparison with the existing state-of-the-art (SOTA) method ConfGF (Shi et al., 2021), we reproduce its data collection and split settings rigorously. Further details are described in Appendix A.3.2.

**Metrics.** Given the RMSD of heavy atoms that measures the distance between generated conformation and the reference, Coverage (COV) and Matching (MAT) scores are defined to measure the diversity and accuracy for a given RMSD threshold $\delta$ respectively.

$$\text{COV}(S_g, S_r) = \frac{1}{|S_r|}|\{R \in S_r | \text{RMSD}(R, \hat{R}) < \delta, \hat{R} \in S_g\}|, \tag{4.4}$$

$$\text{MAT}(S_g, S_r) = \frac{1}{|S_r|} \sum_{R \in S_r} \min_{\hat{R} \in S_g} \text{RMSD}(R, \hat{R}), \tag{4.5}$$

where $S_g$ and $S_r$ denote generated and reference conformations, respectively.

**Implementation Details.** The EVFN is equipped with 4 Graph Transformer blocks and the hidden dimensions are set to 288. All models are trained with Adam optimizer via the loss function (4.3) for 400 epochs. For each molecule in the test set, we follow (Shi et al., 2021) to sample twice as many conformations as the reference ones from each model. We provide all hyperparameters of the score-based framework in Appendix A.3.2.

**Baselines.** We compare EVFN to four classic methods for conformation generation. Specifically, both RDKit (Landrum, 2013) and CGCF(Xu et al., 2021) are distance-based approaches. ConfGF (Shi et al., 2021) is most close to us, attempting to generate conformations by learning the gradient field of the data distribution in an equivariant manner. However, ConfGF only utilizes the distance matrix as the geometric input. We also reproduce EGNN on this task as our baseline.

**Results.** We summarize the mean and median COV and MAT scores on two benchmarks for all methods. As shown in Table 2, EVFN achieves the best performance on almost all metrics and datasets, demonstrating the effectiveness of our proposed method. Compared with ConfGF which employs a similar learning strategy with us, EVFN significantly increases $26.5\%$ COV-mean and $26.6\%$ COV-median scores on the GEOM-Drugs dataset. A potential interpretation is that molecules in Drugs usually contain more atoms and complex chemical functional groups (e.g., Benzene rings) than those of QM9, thus distance-based geometry is not sufficient to model the gradient field of this complex distribution. EVFN also achieves significant improvement in the **Distributions Over Distances** task, and we provide the empirical results and further discussions in Appendix A.3.2.

**Ablations.** Although the superior performance on multiple tasks verifies the effectiveness of EVFN, it remains unclear whether the proposed strategies make a critical contribution. In light of this, we set up several ablative configurations and list the empirical results in Table 3. For the scalarization block, we conduct a variant of EVFN without the scalarization block, named *EVFN w/o Sca.*, which only takes the distance matrix of molecules as the input geometric feature. The results show that including scalarization block plays an important role in the model, as the COV-mean and COV-median scores on the Drugs dataset increase by $23.0\%$ and $21.9\%$, respectively, which implies that including all geometric information will boost the performance of the model. For the graph transformer block,

Table 3: Ablations for EVFN on two datasets.

| Dataset | GEOM-QM9 | | | | GEOM-Drugs | | | |
|---|---|---|---|---|---|---|---|---|
| Metric | COV (%)↑ | | MAT (Å)↓ | | COV (%)↑ | | MAT (Å)↓ | |
| | Mean | Median | Mean | Median | Mean | Median | Mean | Median |
| EVFN $w/o$ Sca | 88.99 | **94.55** | 0.3050 | 0.3066 | 65.67 | 75.63 | 1.1410 | 1.1132 |
| EVFN $w/o$ GT | 85.21 | 91.18 | 0.3057 | 0.3060 | 70.57 | 81.82 | 1.1075 | 1.1004 |
| EVFN | **90.21** | 93.14 | **0.2430** | **0.2457** | **88.64** | **97.56** | **0.9040** | **0.9023** |

we replace the block with the GIN network (Xu et al., 2018) that is employed in ConfGF, getting the variant named *EVFN w/o GT*. The results indicate that introducing the attention mechanism will also contribute to the gradient field modeling. We cannot conduct the ablative study for the vectorization block because it guarantees the output of EVFN is an equivariant vector field.

# 5 RELATED WORK

**Equivariant neural network.** Existing equivariant networks with a theoretical guarantee can be roughly classified into two categories by whether conducting all operations in the original space or not. The first category of methods lift the data into high-dimensional spaces (e.g., lie group) or introduce equivariant functions (e.g., spherical harmonics) to preserve equivariance (Worrall et al., 2017; Thomas et al., 2018; Kondor et al., 2018; Weiler et al., 2018b;a; Weiler & Cesa, 2019; Esteves et al., 2020; Romero et al., 2020; Klicpera et al., 2020; Anderson et al., 2019). They exhibit sufficient expressive power but usually bring expensive computational costs. Our work follows methods of the second category that operates on the original space in a computationally efficient way (Schütt et al., 2018; Köhler et al., 2019; Shi et al., 2021). However, most of these approaches (e.g., EGNN (Satorras et al., 2021c)) abandon a certain amount of geometric (tensor) information, causing their expressive power to be restricted. Different from existing methods, we propose a novel architecture that avoids complex vector-level transformations while preserving complete geometric information (see the discussion at the end of section 3). Essentially, the scalarization is also adopted by EGNN, which is only computed from one basis and EVFN extends the one basis (i.e., radical direction $x_i - x_j$) in EGNN into a tuple of bases for information lossless.

**Gradient fields modeling.** Gradient fields modeling is one of the popular tools for modeling many-body systems from predicting motion trajectories of physical systems (Greydanus et al., 2019; Norcliffe et al., 2020; Li et al., 2020) to estimating probabilistic densities of complex systems (Song & Ermon, 2019; Cai et al., 2020; Shi et al., 2021). Neural ODEs (Chen et al., 2018; Zhuang et al., 2020) are built for learning the gradient of a system by the adjoint method. With the estimated gradient, the NODEs can make predictions for irregular time series by integrating to any given time. Score-based methods attempt to model the data distribution by learning the gradient of its probabilistic density (Song & Ermon, 2020; Song et al., 2020). More precisely, Sohl-Dickstein et al. (2015); Shi et al. (2021); Wu et al. (2021) model the equilibrium states by going through the process determined by the gradient of the density.

# 6 CONCLUSION AND FUTURE WORK

To learn the gradient fields for many-body system modeling, we propose an equivariant vector field neural network (EVFN) aiming at lossless utilization of tensors without incorporating high-dimensional spaces or equivariant functions. With the proposed scalarization technique, EVFN could cooperate with any neural networks without concerns about breaking the equivariance symmetry. Theoretical analyses and extensive empirical results verify the effectiveness of the proposed method. In the future, we will investigate the performance of EVFN in large-scale many-body systems and extend the strategy to other symmetry groups or local symmetry group such as gauge group (Cohen et al., 2019a).

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

# A APPENDIX

## A.1 PROOF AND DISCUSSION ON SCALARIZATION AND VECTORIZATION

**Proposition A.1.** *The complete basis $(\boldsymbol{a}(t), \boldsymbol{b}(t), \boldsymbol{c}(t))$ defined by (3.1) is equivariant under SO(3) transformation of the spatial space.*

*Proof.* Let $g \in SO(3)$, then under the action of $g$, the positions of the many-body system $X(t)$ transform equivariantly:

$$(\boldsymbol{x}_1(t), \ldots, \boldsymbol{x}_n(t)) \xrightarrow{g} (g\boldsymbol{x}_1(t), \ldots, g\boldsymbol{x}_n(t)).$$

Then from the definition of $\boldsymbol{a}(t)$, we know that

$$\boldsymbol{a}(t) \xrightarrow{g} g\boldsymbol{a}(t).$$

For $\boldsymbol{b}(t)$, since

$$(g\boldsymbol{x}_i(t)) \times (g\boldsymbol{x}_j(t)) = \det(g)(g^T)^{-1}(\boldsymbol{x}_i(t) \times \boldsymbol{x}_j(t)) \tag{A.1}$$
$$= g(\boldsymbol{x}_i(t) \times \boldsymbol{x}_j(t)), \tag{A.2}$$

where we have used $g^{-1} = g^T$ for orthogonal matrix $g$ to get the last line. Therefore, $\boldsymbol{b}(t) \xrightarrow{g} g\boldsymbol{b}(t)$. Applying (A.1) once again, we have $\boldsymbol{c}(t) \xrightarrow{g} g\boldsymbol{c}(t)$. $\square$

**Frame bundle and scalarization technique** A frame at $x \in M$ is a linear isomorphism $u$ from $\mathbb{R}^d$ to the tangent space at $x$:$T_x M$. We use $F(M)_x$ to denote the space of all frames at $x$. Then $GL(d, \mathbb{R})$ acts on $F(M)_x$ by

$$\mathbb{R}^d \xrightarrow{g} \mathbb{R}^d \xrightarrow{u} T_x M.$$

Then the frame bundle

$$F(M) = \cup_{x \in M} F(M)_x$$

can be made into a differential manifold. From the principle bundle point of view, each differential manifold $M$ is a quotient of its frame bundle $F(M)$ by the general linear group $GL(d, \mathbb{R})$: $M = F(M)/GL(d, \mathbb{R})$. We denote the quotient map by $\pi$: $F(M) \xrightarrow{\pi} M$. Then, each point of $\boldsymbol{u} \in F(M)$ is a reference frame located at $\boldsymbol{x} := \pi(u) \in M$. On the other hand, $\mathbb{R}^d$ can be seen as a $d$-dimensional differential manifold with a global coordinates chart and SO(3) is the structure-preserving group of the Euclidean metric with a fixed orientation.

Following Hsu (2002), let $\{e_i, 1 \le i \le d\}$ be the canonical basis of $\mathbb{R}^d$, and $\{e^i\}$ the corresponding dual basis. At each frame $u$, the vectors $Y_i := ue_i$ form a basis of $T_x M$. Let $\{Y^i\}$ be the dual frame of $T_x^* M$, then a (r,s)-tensor $\theta$ can be expressed as

$$\theta = \theta_{j_1 \cdots j_s}^{i_1 \cdots i_r} Y_{i_1} \otimes \cdots \otimes Y_{i_r} \otimes Y^{j_1} \otimes \cdots \otimes Y^{j_s}.$$

The **scalarization** of $\theta$ at $u$ is

$$\tilde{\theta}(u) := \theta_{j_1 \cdots j_s}^{i_1 \cdots i_r} e_{i_1} \otimes \cdots \otimes e_{i_r} \otimes e^{j_1} \otimes \cdots \otimes e^{j_s}. \tag{A.3}$$

Through scalarization, a tensor field $\theta$ becomes an ordinary vector space valued function on $F(M)$:

$$\tilde{\theta} : F(M) \to \mathbb{R}^r \otimes \mathbb{R}^s.$$

Therefore, geometric operations such as covariant derivative and tensor product on manifolds can be realized as directional derivative and tensor product of ordinary vector spaces (Hsu, 2002).

**Proposition A.2.** *There is a one-to-one correspondence between the scalarization of tensor fields on the orthonormal frame bundle $O(\mathbb{R}^3)$ (A.3) and the SO(3)-invariant scalars tuple obtained by the scalarization block (3.4) under an equivariant basis.*

*Proof.* Since we are working in $\mathbb{R}^3$ with a fixed orientation, the $GL(3, \mathbb{R})$ group action is reduced to $SO(3)$ global group action acting on the orthonormal frame bundle $O(\mathbb{R}^3)$. A frame $u \in O(\mathbb{R}^3)$ at $\pi(u) \in \mathbb{R}^3$ can be transported to another point in $\mathbb{R}^3$ by translation. Therefore, we neglect the

origin of the frame in the proof. Let $u_e = (\boldsymbol{e}_1, \boldsymbol{e}_2, \boldsymbol{e}_3)$ be an equivariant basis of $\mathbb{R}^3$, then given a scalars tuple $\{\theta^{i_1,\ldots,i_r}\}$, we construct a vector-valued function on $O(\mathbb{R}^3)$ by:

$$\tilde{\theta}^{i_1,\ldots,i_r}(u) = g_{i_1 i_1'} \cdots g_{i_r i_r'} \theta^{i_1',\ldots,i_r'},$$

where $g$ is the $SO(3)$ transformation from $u_e$ to another frame $u$. Moreover, $\tilde{\theta}^{i_1,\ldots,i_r}(u)$ is SO(3)-equivariant, since

$$\tilde{\theta}(gu) = g\tilde{\theta}(u),$$

where the $g$ on the right side means the usual extension of the action of SO(3) from $\mathbb{R}$ to the tensor space $\mathbb{R}^{\otimes r}$. We have constructed the (r,0) tensor field from the scalars tuple. It's easy to check that $\tilde{\theta}^{i_1,\ldots,i_r}(u)$ induces a (r,0) tensor field on $\mathbb{R}^3$ by following the definition of (2.1).

From scalarization $\tilde{\theta}^{i_1,\ldots,i_r}(u)$ on $O(\mathbb{R}^3)$ to scalarization is obvious. Note that any equivariant basis $u_e$ is also a point on $O(\mathbb{R}^3)$, therefore the scalars tuple is just the values of $\tilde{\theta}^{i_1,\ldots,i_r}$ at $u_e$:

$$\theta^{i_1,\ldots,i_r} = \tilde{\theta}^{i_1,\ldots,i_r}(u_e).$$

For general (r,s)-type tensors, the proof is the same by adding the dual basis of $u_e$. □

**Remark A.3.** *For molecular structures, one common obstruction for distance-based modeling (Shi et al., 2021) is the chirality problem (Bruice, 2000). Therefore it's meaningful to investigate how the equivariant basis (3.1) transforms under reflection $\boldsymbol{x} \to -\boldsymbol{x}$. Notice that $\boldsymbol{a} \to -\boldsymbol{a}$, and*

$$\boldsymbol{b} = \boldsymbol{x}_i \times \boldsymbol{x}_j \to \boldsymbol{b}.$$

*It implies that $\boldsymbol{c} \to -\boldsymbol{c}$. In conclusion, the orientation of the equivariant basis remains unchanged under reflection.*

### A.2 METHODOLOGY

#### A.2.1 PERMUTATION AND TRANSLATION EQUIVARIANCE

For a many-body system $\boldsymbol{X}(t) = (\boldsymbol{x}_1(t), \ldots, \boldsymbol{x}_n(t))$, the centroid $C(t)$ is defined by

$$\boldsymbol{c}(t) = \frac{\boldsymbol{x}_1(t) + \cdots \boldsymbol{x}_n(t)}{n}.$$

Translating the reference by $\boldsymbol{h}$, then

$$X(t) + h \to \boldsymbol{c}(t) + h.$$

Therefore, recentering the reference's origin to the centroid at the input's time $t = 0$, we have

$$\boldsymbol{X}(t) - \boldsymbol{c}(0) \xrightarrow{\text{translation by } \boldsymbol{h} \text{ at } t=0} \boldsymbol{X}(t) - \boldsymbol{c}(0).$$

That is, the system is translation-invariant under the recentered reference if the translation is done at the input's time $t = 0$, which is exactly the scenario considered in predicting the future trajectory or state. Note that although the gradient field is supposed to be translation-invariant, the state of the many-body system after integrating should be translation-equivariant. Therefore, we add $\boldsymbol{c}(0)$ back for the output of the evolving block.

To emphasis the permutation symmetry, we quote the Kolmogorov–Arnold representation theorem: if $f(\boldsymbol{x}_1, \ldots, \boldsymbol{x}_n)$ is a permutation invariant multivariate continuous function, then

$$f(\boldsymbol{x}_1, \ldots, \boldsymbol{x}_n) = g(\sum_{i=1}^{n} \phi(\boldsymbol{x}_i)). \tag{A.4}$$

The crucial point is the function $\phi$ is shared among all the points, which exactly fits the definition of the so-called message-passing scheme. For a many-body system $X$, let $v = (v_1, \ldots v_n)$ be its vector field, then $v_i \in \mathbb{R}^3$ corresponds the equivariant vector for $\boldsymbol{x}_i \in \{\boldsymbol{x}_1, \ldots, \boldsymbol{x}_n\}$. Denote EVFN with parameters $\theta$ by $\Phi^{\theta,t} = \{\Phi_i^\theta\}_{i=1}^n$, then

$$v_i(t) = \Phi_i^\theta(\boldsymbol{X}(t), t),$$

for a fixed particle $\boldsymbol{x}_i$. Suppose $(i_1, \ldots, i_k)$ (neighbors of $\boldsymbol{x}_i$) indicate indexes of particles which have interaction with $\boldsymbol{x}_i$, then obviously $1 \leq k \leq n - 1$. By (A.4), $\Phi_i(\boldsymbol{X}(t), t)$ is an aggregation of message from $\boldsymbol{x}_i$'s neighbors, therefore we have:

$$\Phi_i(X(t), t) = \frac{1}{k} \sum_{j=1}^{k} \phi(\boldsymbol{x}_i(t), \boldsymbol{x}_{i_j}(t), t), \tag{A.5}$$

and $\phi$ is a SO(3)-equivariant network with vector output. Note that (A.5) performs aggregation at the level of vectors, therefore we choose $g$ in (A.4) to be the arithmetic mean to preserve SO(3) symmetry. A **shared** neural network $\phi$ guarantees the permutation equivariance of EVFN.

### A.2.2 GRAPH TRANSFORMER BLOCK

For a many-body system $\boldsymbol{X}$, we first represent it as a spatial graph and utilize an attention-based to learn the SO(3)-invariant edgewise embeddings $m_{ij}$ from the graph. Considering that there does not exist any graph topology in most real-world scenarios, we introduce the attention mechanism due to its powerful capacity in learning the correlations between inter-instances (Vaswani et al., 2017). Now we discuss the workflow of the GTB block. Note that here we omit the time index of the SO(3)-invariant scalars for the simplicity of description.

**Feature embedding**   Given geometric scalars $t_{ij}$, node features $h_i$ and edge features $m_{ij}$, GTB first embeds them into high-dimensional representations:

$$h_i = \text{MLP}(h_i), \tag{A.6}$$
$$e_{ij} = \text{MLP}(e_{ij}), \tag{A.7}$$
$$t_{ij} = \text{Fourier}(t_{ij}), \tag{A.8}$$
$$e_{ij} = e_{ij} + t_{ij}, \tag{A.9}$$

where MLP denotes a fully connected network and Fourier denotes a Fourier transformation with a tuple of learnable frequencies.

**Transformer block**   The overall architecture of our GTB block is inspired by (Shi et al., 2020). For each GTB block, we first compute the edge-wise message with $\phi_m^1$ (A.7), then leverage the message embeddings and node embeddings with a transformer encoder-like architecture to refine the node embeddings. After that, we update edgewise messages with a residue block (A.13). The whole pipeline is expressed as (A.9)-(A.15).

$$m_{ij} = \phi_m^1(h_i, h_j, e_{ij}), \tag{A.10}$$
$$q_i = \phi_q(h_i), k_{ij} = \phi_k(h_i, m_{ij}), v_{ij} = \phi_v(m_{ij}), \tag{A.11}$$
$$\alpha_{ij} = \frac{\langle q_i, k_{ij} \rangle}{\sum_{j' \in \mathcal{N}(i)} \langle q_i, k_{ij'} \rangle}, \tag{A.12}$$
$$\mathcal{M}_i = \text{LayerNorm}\Big( \sum_{j \in \mathcal{N}(i)} \alpha_{ij} v_{ij} \Big), \tag{A.13}$$
$$h_i = \phi_h(h_i, \mathcal{M}_i), \tag{A.14}$$
$$h_i = h_i + \text{LayerNorm}(h_i), \tag{A.15}$$
$$m_{ij} = \phi_m^2(h_i, h_j, e_{ij}) + m_{ij}, \tag{A.16}$$

where $\phi_m^1$, $\phi_q$, $\phi_k$, $\phi_v$, $\phi_h$ and $\phi_m^2$ are fully connected networks. $\alpha_{ij}$ and $\mathcal{M}_i$ denote the attention weights and the refined nodewise embeddings, respectively. LayerNorm refers to the normalization layer adopted in (Vaswani et al., 2017).

### A.2.3 EQUIVARIANCE OF EVFN

We need the following lemma:

**Lemma A.4.** *Suppose $h$ is an invariant function, $f_1, \ldots, f_k$ are arbitrary nonlinear functions. Then, the composition $f_k \circ \cdots \circ f_1 \circ h$ is an invariant function.*

*Proof.* Consider the group action $g$ acting on $x \in \mathbb{R}^3$, then $f$ is invariant means that $f \circ g(x) := f(gx) = f(x)$. We have

$$f_k \circ \cdots \circ f_1 \circ g = f_k \circ \cdots \circ f_1,$$

hence the composition $f_k \circ \cdots \circ f_1$ is invariant. $\qquad\square$

Let $f_1$ be the scalarization block. Because the output of $f_1$ are scalars (from (3.3)) and SO(3)-scalars must be invariant under SO(3) group action, we can conclude that $f_1$ is invariant. Denote each graph-transformer layer by $f_i$, $i \in \{2, \ldots, k\}$, then by Lemma A.4, we can conclude that the output of the graph-transformer block is also invariant.

Finally, from the standard fact that scalars multiply vectors yield equivariant vectors, we conclude that the output of the vectorization block is SO(3)-equivariant. Combining with the translation equivariance in A.2.1, our model is SE(3)-equivariant.

### A.2.4 DIHEDRAL ANGLE

For a given node $\boldsymbol{x}_l$ with three neighbors $\boldsymbol{x}_i$, $\boldsymbol{x}_j$ and $\boldsymbol{x}_k$. The dihedral angle of the plane spanned by $(\boldsymbol{x}_l - \boldsymbol{x}_i, \boldsymbol{x}_l - \boldsymbol{x}_j)$ and the plane spanned by $(\boldsymbol{x}_l - \boldsymbol{x}_i, \boldsymbol{x}_l - \boldsymbol{x}_k)$ is given by the inner product of normal vectors of the two planes:

$$\frac{(\boldsymbol{x}_l - \boldsymbol{x}_i) \times (\boldsymbol{x}_l - \boldsymbol{x}_j)}{\|(\boldsymbol{x}_l - \boldsymbol{x}_i) \times (\boldsymbol{x}_l - \boldsymbol{x}_j)\|},$$

and

$$\frac{(\boldsymbol{x}_l - \boldsymbol{x}_i) \times (\boldsymbol{x}_l - \boldsymbol{x}_k)}{\|(\boldsymbol{x}_l - \boldsymbol{x}_i) \times (\boldsymbol{x}_l - \boldsymbol{x}_k)\|}.$$

Therefore, the dihedral angle is a function of the position vectors $\boldsymbol{x}_l$, $\boldsymbol{x}_i$, $\boldsymbol{x}_j$ and $\boldsymbol{x}_k$, which cannot be determined purely by the positions' norm: $\|\boldsymbol{x}_l - \boldsymbol{x}_i\|, \|\boldsymbol{x}_l - \boldsymbol{x}_j\|$ and $\|\boldsymbol{x}_l - \boldsymbol{x}_k\|$.

However, by the complete equivariant basis, the position vectors are fully expressed (only the coordinates are transformed from the natural coordinate system to the equivariant coordinate system), then all functions of the position vectors can also be expressed.

### A.2.5 EVOLVING BLOCK

According to the order of vector field in dynamic systems, we design different evolving blocks to integrate the estimated vector field to obtain the dynamics.

Recall that given a multi-particle Newton dynamic system $X(t) = \{\boldsymbol{x}_i(t) \in \mathbb{R}^3, i = 1, 2, \cdots, n\}$ whose underlying physical rules are unknown, let $\boldsymbol{v}_i$ denote the velocity of particle $\boldsymbol{x}_i$. We attempt to predict its position or trajectory by integrating a second-order equivariant vector field. In this case, the EVFN network $\Phi$ is implemented for modeling the acceleration vector $\dot{\boldsymbol{v}}(t)$. More precisely, the hidden differential unit of NODE (Chen et al. (2018)) has the following form:

$$\begin{bmatrix} \dot{X}(t) \\ \dot{\boldsymbol{v}}(t) \end{bmatrix} = \begin{bmatrix} \boldsymbol{v}(t) \\ \Phi(X(t), \boldsymbol{v}(t), t) \end{bmatrix}. \tag{A.17}$$

For the $i$-th particle,

$$\dot{\boldsymbol{v}}_i(t) = \Phi_i(X(t), \boldsymbol{v}(t)) := \sum_{j=1}^{k} \phi(\boldsymbol{x}_i(t), \boldsymbol{x}_{i_j}(t), t). \tag{A.18}$$

The ordinary differential equation (A.17) is then solved by black-box ODE solver as in (Chen et al., 2018). We denote the evolving block in the second-order vector field case by **SNODE**.

As to the statistical ensemble system, we try to predict the reverse evolving process from a random state to equilibrium by integrating a first-order equivariant vector field. For example, all physical-allowable molecule conformations are located in an equilibrium state determined by the energy function. Suppose the forward process from equilibrium to non-equilibrium of the system satisfies:

$$d\boldsymbol{X}(t) = f(\boldsymbol{X}(t), t)dt + g(t)dW_t, \ \ 0 \le t \le T,$$

where $W_t$ is the Brownian motion and the initial state $\boldsymbol{X}(0)$ follows an unknown equilibrium distribution $p_0$. Denote the marginal distribution at time $t$ by $p_t$, then

$$p_t(\boldsymbol{X}) = \exp\{-\beta \boldsymbol{H}_t(X)\},$$

so the Hamiltonian function $\boldsymbol{H}_t$ at time $t$ is entangled with $p_0$. According to the Liouville equation (the probability flow in Song et al. (2020)), the reverse evolving process satisfies the following ODE:

$$d\boldsymbol{X}(T-t) = f(\boldsymbol{X}(T-t), T-t)dt - \frac{1}{2}g^2(T-t)\nabla \boldsymbol{H}_{T-t}(\boldsymbol{x}_{T-t})dt.$$

The gradient field of the Hamiltonian function are also called the force field. Therefore the NODE evolving block has the following form:

$$\dot{\boldsymbol{X}}(T-t) = \Phi(\boldsymbol{X}(T-t)), \ \ 0 \le t \le T.$$

and for the $i$-th particle, following form:

$$\Phi_i(\boldsymbol{X}(T-t)) = f_i(\boldsymbol{X}(T-t), T-t) - \frac{1}{2}g^2(T-t)[\sum_{j=1}^{k}\phi(\boldsymbol{x}_i(T-t), \boldsymbol{x}_{i_j}(T-t), T-t)]. \ \ \text{(A.19)}$$

In this case, we will use the PC scheme proposed by (Song et al., 2020) for correcting the numerical integration error. Note that the vector-valued function $f(x,t)$ and the scalar function $g(t)$ is prior knowledge and set to be fixed, so the only learnable module in EVFN is the vector field network $\phi$. In Shi et al. (2021) and our molecular experiment, we use the discretization of VP SDE (Song et al., 2020), where $f \equiv 0$ and

$$g(t) = \sqrt{\frac{[d\sigma^2(t)]}{dt}}.$$

### A.2.6 Neural ODE as a continuous limit

The idea of neural ODE is to fit dynamics by modelling its infinitesimal rates of change (gradient) and implementing numerical integration. The whole process can be seen as taking the continuous limit of the residual network as the depth goes to infinity. Consider a residual network where all hidden layers have the same dimension, denote the neural network's parameters at t-th layer by $\theta_t$, then

$$h_{t+1} = h_t + f(h_t, \theta_t), \ \ t \in \mathbb{N}^+.$$

Taking the discrete layer index $t$ to its continuous limit, we get

$$\frac{dh(t)}{dt} = f(h(t), \theta_t), \ \ t \in \mathbb{R}^+.$$

The discrete back-propagation method also has a continuous limit: the adjoint method Chen et al. (2018); Norcliffe et al. (2020).

### A.2.7 Scalability

Compared to EGNN, the extra cost of EVFN is to calculate the values of the scalars in $t_{ij}, \forall i, j$. Compared to the computational cost of back-propagation in the neural networks, the cost of calculating the $O(3 * 3 * N^2)$ (N is the number of nodes in the graph) scalars is much less. For example, for the molecular conformation generation task, transforming the tensors into SO(3)-invariant scalars only brings 9.6% extra real-time computational cost and 17.4% extra memory cost.

### A.3 Experiment

### A.3.1 Newtonian many-body system

**Partially observed system** This system consists of six particles under Newton's gravitation but only four of them could be observed, i.e., for each trajectory we are provided with positions $\boldsymbol{X}(t) \in \mathbb{R}^{4 \times 3}$ and velocities $\boldsymbol{V}(t) \in \mathbb{R}^{4 \times 3}$. The time evolution of the particles is given by

$$\ddot{\boldsymbol{x}}_i = \sum_{j \in \{1,...,i-1,i+1,...6\}} -m_j \frac{\boldsymbol{x}_i - \boldsymbol{x}_j}{\|\boldsymbol{x}_i - \boldsymbol{x}_j\|^3}, \ \ 1 \le i \le 4. \ \ \text{(A.20)}$$

**Gravity field**  This system consists of six particles under both the mutual newton's gravitation and an external static force of the form: $\boldsymbol{f}_g = (0,\,0,\,\eta)$. The time evolution of the particles are given by

$$\ddot{\boldsymbol{x}}_i = \sum_{j \in \{1,\ldots,i-1,i+1,\ldots6\}} -m_j \frac{\boldsymbol{x}_i - \boldsymbol{x}_j}{\|\boldsymbol{x}_i - \boldsymbol{x}_j\|^3} + \boldsymbol{f}_\eta, \quad 1 \le i \le 6. \tag{A.21}$$

**Lorentz force field**  This system consists of three particles and controlled by Newton's law of motion and a Lorentz field, which means there exists a force field perpendicular to the direction of velocity $\boldsymbol{v}$, i.e., $\boldsymbol{f}_l(\boldsymbol{v}) = q\boldsymbol{v} \times \mathbf{B}$, where $q$ and $\mathbf{B}$ denote the charge of particles and the direction vector of the electromagnetic field respectively. The time evolution of the particles is given by:

$$\ddot{\boldsymbol{x}}_i = \sum_{j \in \{1,\ldots,i-1,i+1,\ldots3\}} -m_j \frac{\boldsymbol{x}_i - \boldsymbol{x}_j}{\|\boldsymbol{x}_i - \boldsymbol{x}_j\|^3} + \boldsymbol{f}_l^i(\boldsymbol{v}_i), \quad 1 \le i \le 3. \tag{A.22}$$

**Implementation Details**  Following (Zhuang et al., 2020), all trajectories are simulated using the *Dopri5* solver (Dormand & Prince, 1980) with the tolerance to $10^{-7}$ and the modified physical rules. The trajectory points are uniformly sampled with $\Delta t = 10^{-3}$ for the POS dataset or $\Delta t = 5 \cdot 10^{-4}$ for the other two datasets. The number of training labels (i.e., trajectory points) for three tasks is 1.5k, 100k and 100k, respectively. We implement all baselines and our method with Pytorch (Paszke et al., 2019). All models use the same ODE solver (*Dopri5*) as the evolving blocks and are trained with Adam optimizer (Kingma & Ba, 2014) via an MSE loss for 800 epochs. We set the number of layers to 2 for all models and adjust the hidden dimensions of each model separately to keep the parameters in the same level. We adopt EGNN from (Satorras et al., 2021a) for outputting vectors. For all datasets, we take the time $t$ and the Ł$_2$ norm of velocities as the node features and take the relative distances as the edge features.

**Further Analysis of Results**  As shown in Table 1, the non-equivariant GCN performs the best with the original input on SEFF task. The reason is that the equivariant models are superior to model the force that is a function of the radial direction, however, the external force in SEFF is a constant vector that is independent with the radial direction.

We also compare EVFN with the non-computational efficient equivariant models such as SE(3)-Transformer (Fuchs et al., 2020), which leverages spherical harmonics to preserve equivariance. In practice, we find that it is hard to jointly optimize the neural ODE and the SE(3)-Transformer model by directly combining the two models together. And the empirical results in Table 4 show that SE(3)-Transformer achieves an extremely poor performance compared with EVFN. To the best of our knowledge, there is no previous work studying the combination of SE(3)-Transformer and neural ODEs. We would like to further explore this problem in future work. For the results in Table 4, all SE(3)-Transformer models are implemented according to the official codebase [5], where the number of hidden layers is set to 2 and each layer has representation degrees $\{0, 1\}$ and channels $\{8, 4\}$. This setting makes the number of parameters of SE(3)-Transformer (i.e., 40k) comparable with EVFN and other baselines. We also implemented another variant with the default setting of SE(3)-Transformer, where each layer has representation degrees $\{0, 1, 2, 3\}$ and 8 channels per degree. We find it would bring much more parameters (i.e., 118k) and computational efforts, but only marginal improvement (e.g., 0.405 vs. 0.422 for MSE on the POS dataset).

**Data Augmentation**  For all Newtonian many-body systems, we propose to force a naive GCN to be SO(3) equivariant by introducing numerous SO(3) augmentation samples. The augmentation strategy is that, for each iteration, we rotate the initial condition and the ground-truth trajectory with a random probability p. We visualize four MSE metrics of GCN (Aug) with different augmentation probabilities on the POS dataset in Figure 2. As shown in Figure 2, with the original input, the interpolation and extrapolation capacity of GCN is damaged as p grows. with the rotated input, the MSE of GCN (Aug) gradually decreases as p grows, but it is still higher several magnitudes than that of real equivariant networks, implying that it is difficult to achieve equivariance with simple data augmentation.

---

[5]https://github.com/FabianFuchsML/se3-transformer-public

Table 4: Comparison with SE(3)-Transformer

| Setting | Method | Inter. | Extra. | Rot. Inter. | Rot. Extra. | $\Delta_{EQ}$ |
|---------|--------|--------|--------|-------------|-------------|---------------|
| POS | SE(3)-Transformer | 0.422 | 4.555 | 0.442 | 4.559 | $7.82 \cdot 10^{-4}$ |
| | EVFN | 0.138 | 2.502 | 0.137 | 2.428 | $1.23 \cdot 10^{-4}$ |
| SEFF | SE(3)-Transformer | 12.623 | 53.358 | 12.623 | 53.258 | $3.67 \cdot 10^{-5}$ |
| | EVFN | 0.139 | 1.289 | 0.142 | 1.310 | $1.05 \cdot 10^{-6}$ |
| DEFF | SE(3)-Transformer | 9.119 | 37.313 | 9.119 | 37.312 | $3.31 \cdot 10^{-4}$ |
| | EVFN | $1.38 \cdot 10^{-3}$ | $2.43 \cdot 10^{-2}$ | $1.39 \cdot 10^{-3}$ | $2.48 \cdot 10^{-2}$ | $1.15 \cdot 10^{-5}$ |

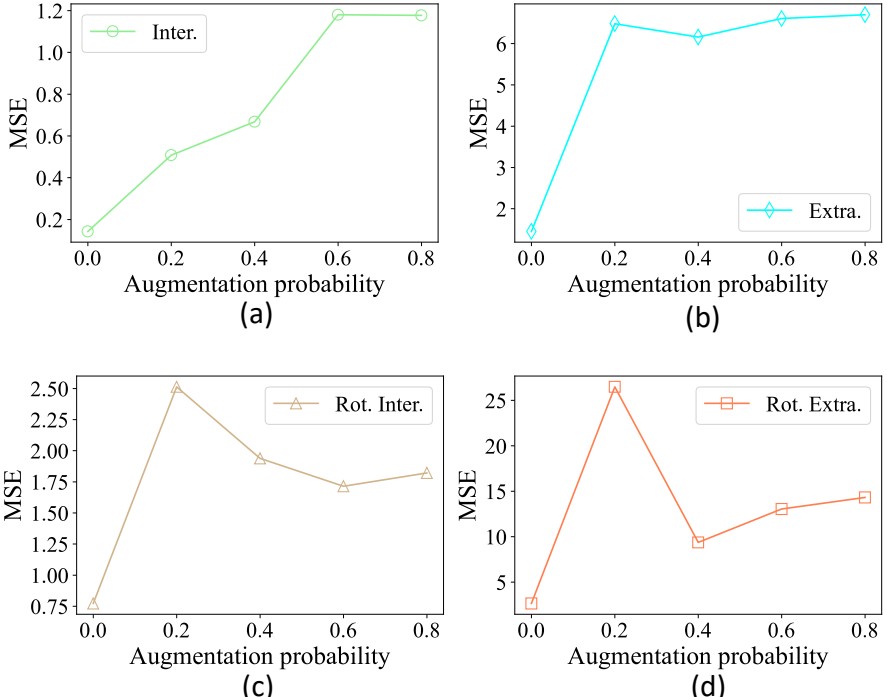

Figure 2: Results of GCN with various data augmentation degrees on the POS dataset

**The Impact of the Number of Training samples** To further explore the impact of the number of training data for both non-equivariant and equivariant models, we re-train GCN and EVFN with the different number of training samples on the POS dataset and summarize the results in Figure 3. In particular, under the same data splitting setting described in Section 4.1, we choose to sample the trajectories with different time intervals $\{10^{-2}, 1.5 \cdot 10^{-3}, 5 \cdot 10^{-4}, 3 \cdot 10^{-4}\}$, corresponding to 150, 1000, 3000 and 5000 training labels, respectively. The time interval of the test set remains the same as before (i.e., $10^{-3}$). Note that we avoid introducing new trajectories into the dataset because it may influence the problem complexity of the original interpolation and extrapolation tasks. As shown in Figure 3 (a) and (b), with the number of samples growing, the interpolation performance of GCN becomes better but its extrapolation becomes worse, indicating that the GCN-based neural ODE may easily overfit to the interpolation task. Compared to GCN, EVFN achieves more robust performance to the different number of training samples. As shown in Figure 3 (c) and (d), introducing more training samples in the original coordinate frame cannot enhance the equivariant capacity of GCN efficiently, which demonstrates that SO(3) augmentation is a more reasonable choice to realize equivariance for non-equivariant methods.

**Equivariant Errors** As discussed in Section 4.1, the bigger equivariance error of EVFN (Table 1, col 7) is caused by numerical errors introduced by the basis construction procedure and the neural

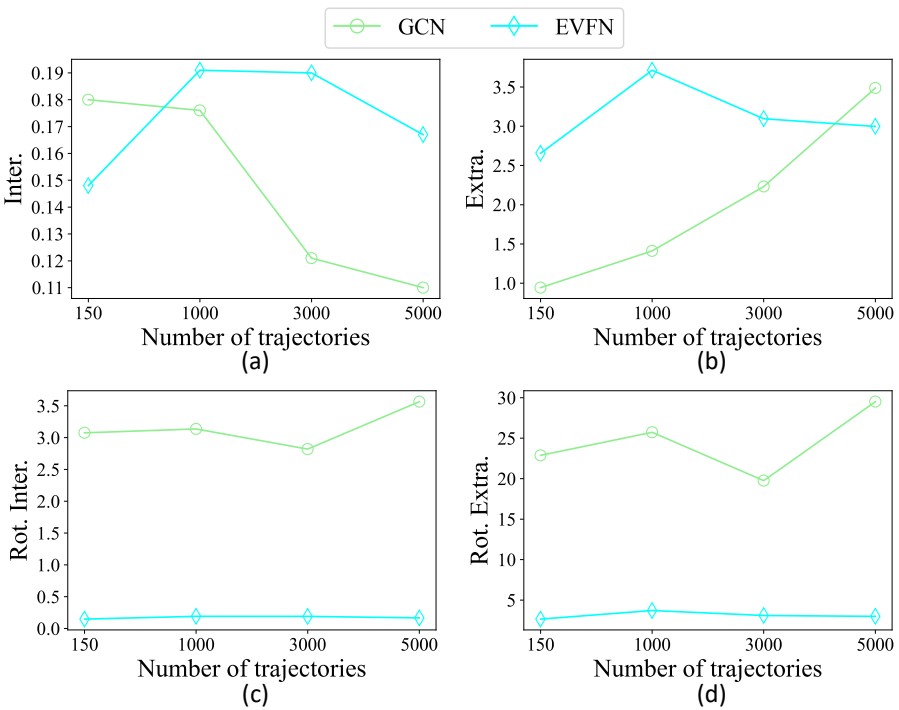

Figure 3: Results for different number of training trajectories on the POS dataset

Table 5: The impact of different normalization constants. SE(3)-T denotes the SE(3)-Transformer model.

| Setting | Method | $C$ | Inter. | Extra. | Rot. Inter. | Rot. Extra. | $\Delta_{EQ}$ |
|---------|--------|-----|--------|--------|-------------|-------------|---------------|
| POS | EGNN | $\mathcal{O}(1)$ | 0.726 | 6.449 | 0.726 | 6.462 | $9.19 \cdot 10^{-7}$ |
| | SE(3)-T | - | 0.422 | 4.555 | 0.442 | 4.559 | $7.82 \cdot 10^{-4}$ |
| | EVFN | $\mathcal{O}(10^{-3})$ | 0.138 | 2.502 | 0.137 | 2.428 | $1.23 \cdot 10^{-4}$ |
| | EVFN | $\mathcal{O}(1)$ | 0.206 | 2.432 | 0.206 | 2.442 | $4.13 \cdot 10^{-6}$ |
| | EVFN | $\mathcal{O}(10)$ | 0.608 | 2.344 | 0.608 | 2.344 | $3.60 \cdot 10^{-6}$ |
| SEFF | EGNN | $\mathcal{O}(1)$ | 0.463 | 4.248 | 0.467 | 4.275 | $5.88 \cdot 10^{-6}$ |
| | SE(3)-T | - | 12.623 | 53.358 | 12.623 | 53.258 | $3.67 \cdot 10^{-5}$ |
| | EVFN | $\mathcal{O}(10^{-3})$ | 0.139 | 1.289 | 0.142 | 1.310 | $1.05 \cdot 10^{-6}$ |
| | EVFN | $\mathcal{O}(1)$ | 0.213 | 1.888 | 0.213 | 1.888 | $3.38 \cdot 10^{-6}$ |
| | EVFN | $\mathcal{O}(10)$ | 0.498 | 4.417 | 0.496 | 4.414 | $5.41 \cdot 10^{-7}$ |
| DEFF | EGNN | $\mathcal{O}(1)$ | $1.99 \cdot 10^{-2}$ | 0.213 | $3.22 \cdot 10^{-2}$ | 0.290 | $8.55 \cdot 10^{-7}$ |
| | SE(3)-T | - | 9.119 | 37.313 | 9.119 | 37.312 | $3.31 \cdot 10^{-4}$ |
| | EVFN | $\mathcal{O}(10^{-3})$ | $1.38 \cdot 10^{-3}$ | $2.43 \cdot 10^{-2}$ | $1.39 \cdot 10^{-3}$ | $2.48 \cdot 10^{-2}$ | $1.15 \cdot 10^{-5}$ |
| | EVFN | $\mathcal{O}(1)$ | $1.88 \cdot 10^{-3}$ | $3.17 \cdot 10^{-2}$ | $1.88 \cdot 10^{-3}$ | $3.29 \cdot 10^{-2}$ | $2.11 \cdot 10^{-6}$ |
| | EVFN | $\mathcal{O}(10)$ | $1.66 \cdot 10^{-3}$ | $3.13 \cdot 10^{-2}$ | $1.81 \cdot 10^{-3}$ | $3.24 \cdot 10^{-2}$ | $6.51 \cdot 10^{-7}$ |

ODE training framework. More precisely, to build the equivariant basis, we incorporated the normalization operation $\frac{\boldsymbol{x}}{\|\boldsymbol{x}\|} \to \frac{\boldsymbol{x}}{\|\boldsymbol{x}\|+C}$ ($x$ denotes vectors and $C$ denotes the normalization constant) to obtain unit vectors, where the division operation would introduce numerical errors and neural ODE would lead to a larger cumulative error with the integration process. The similar phenomenon is also observed in SE(3)-Transformer (Fuchs et al., 2020) and EGNN (Satorras et al., 2021b). As shown in Table 5, with some suitable normalization constants, EVFN can achieve comparable or even better equivariant errors than other equivariant models, validating the equivariance of the pro-

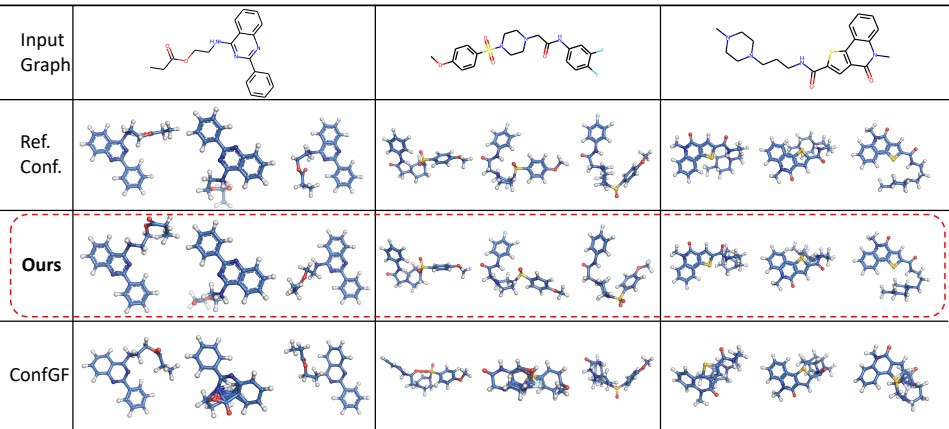

Figure 4: Visualizations of generated conformations. For each molecule randomly selected from GEOM-Drugs dataset, we sample multiple conformations and show the best-aligned ones with the reference ones.

posed method. In most cases, EVFN with larger normalization constants can still outperform other baselines significantly.

### A.3.2 MOLECULAR CONFORMATION GENERATION

**Dataset**  For each dataset, $40,000$ molecules are randomly drawn and $5$ most likely conformations (sorted by energy) are selected for each molecule, and $200$ molecules are drawn from the remaining data, which results in $200,000$ conformations in the training set, $22,408$ and $14,324$ conformations in the test set for GEOM-QM9 and GEOM-Drugs datasets, respectively. The distances over distributions task are evaluated on the ISO17 dataset, where we follow the setup in (Simm & Hernández-Lobato, 2019).

**Implementation Details**  Besides the geometric input, we feed the node type, edge type and relative distances as extra node/edge attributes into the graph transformer block. Our score-based training framework is adapted from (Shi et al., 2021). The maximum and minimum noise scales are set to $10$ and $0.01$. Let $\{\sigma_i\}_{i=1}^{L}$ be a positive geometric progression scheme with a common ratio, we split the noise range into $50$ levels. For the reverse process, we find the performance difference between the ALD sampler and the PC sampler is marginal, so we do not compare this point with quantitative results. We choose the PC sampler as our evolving block and respectively set the iteration steps of predictor and corrector to $10$ and $100$. The sample step size $\eta_s$ is chosen according to (Song & Ermon, 2020). We keep all hyper-parameters mentioned in the forward and reverse process the same as (Shi et al., 2021). The results reported in Table 2 are copied from (Shi et al., 2021) considering that we rigorously evaluate EVFN on the same benchmark and data split setting.

**Conformation Generation**  Here we introduce the calculation equation of RMSD:

$$\text{RMSD}(R, \hat{R}) = \min(\frac{1}{n} \sum_{i=1}^{n} ||R_i - \hat{R}_i||^2)^{\frac{1}{2}}, \tag{A.23}$$

where $n$ denotes the number of heavy atoms.

We visualize several conformations in the Drugs dataset in Figure 4 that are best aligned with the reference ones generated by different methods, illustrating EVFN's superior capacity on generating high-quality drug molecular conformation.

**Distributions over Distances**  To evaluate the distribution of the generated conformations, we utilize maximum mean discrepancy (MMD) (Gretton et al., 2012) to measure the discrepancy between the generated distributions and the reference distributions. As shown in Table 6, EVFN dramatically

Table 6: Accuracy of the distributions over distances generated by different approaches compared to the ground-truth.

| Method | Single | | Pair | | All | |
|--------|--------|--------|--------|--------|--------|--------|
|        | Mean   | Median | Mean   | Median | Mean   | Median |
| RDKit  | 3.4513 | 3.1602 | 3.8452 | 3.6287 | 4.0866 | 3.7519 |
| CGCF   | 0.4490 | 0.1786 | 0.5509 | 0.2734 | 0.8703 | 0.4447 |
| ConfGF | 0.3684 | 0.2358 | 0.4582 | 0.3206 | 0.6091 | 0.4240 |
| **EVFN** | **0.1317** | **0.0420** | **0.1787** | **0.0695** | **0.3185** | **0.1142** |

outperforms the previous SOTA (ConfGF), demonstrating the strong capacity of the proposed model in modeling molecular dynamics data. In particular, EVFN reduces the MMD by a magnitude in both Single-median and Pair-Median metrics.

