# OpenReview forum: "Equivariant Vector Field Network for Many-body System Modeling"
_ICLR.cc/2022/Conference — ICLR 2022 Submitted_

### Official Review · Reviewer_gmFP · 2021-10-29

**Correctness:** 3
**Technical Novelty And Significance:** 2
**Empirical Novelty And Significance:** 2
**Recommendation:** 3
**Confidence:** 4

**Main Review:**

Transforming particle interaction into a rotationally invariant reference frames is a common approach in molecular mechanics, usually achieved with inner coordinates such as distances, angles, dihedral angles etc. These have the disadvantage that some coordinate information is lost, however they fulfill all equivariance / invariance properties w.r.t. rotation, reflection, translation and permutation in local reference frames.

The authors claim that "EVFN is able to represent tensor information losslessly". This is also true for previous equivariant approaches such as TFN, EGNN, PaiNN or NequIP, as long as no distance cutoff is used (see for example Dym and Maron 2020). However, in practice the distance cutoff is essential not only for computational efficiency but also to facilitate generalization across local environments. This is not possible in the proposed approach:
The EVFN transform is rotationally equivariant and translational invariant only with respect to global transforms. This has serious consequences for its capability to generalize, in particular for larger systems. As an extreme example, take two molecules (or clouds of Newtonian particles) at a large enough distance such that the inter-molecular interactions are negligible. Since the proposed coordinate is not translational invariant, the pairwise representations are not invariant to rotating the two molecules separately. Therefore, the proposed representation is not suitable to model general many-particle systems.

Despite these flaws, the method shows good performance on the toy systems and conformer search. A reason might be that for small systems generalization is not the deciding factor, and the baseline methods do not retain the necessary information. To better judge this trade-off, it would be helpful to apply the proposed approach to the related task of predicting molecular force fields (e.g. on the MD17 dataset where a lot of baselines are available for comparison).

Further comments:
- Centering the system prevents application of the method to a box with periodic boundary conditions. Are there any ideas how to overcome this?
- The background section could be made more accessible for readers not familiar with group theory and tensor algebra, since this part is not really necessary to understand the method and proofs in Eq. 3.1-3.3.
- Sec 4.2: the ISO17 dataset was introduced by Schütt et al (2017)
- Why are the equivariance errors in Table 1 slightly higher for EVFN than for the other equivariant methods?

**Summary Of The Paper:**

The paper proposes an rotationally equivariant neural network based on a transform of particle pairs into a rotationally invariant reference frame. This is applied to trajectory prediction of many-body particle systems and molecular conformer search.

**Summary Of The Review:**

The proposed transformation has the flaws outlined above which might prevent generalization across local environments. More experiments are required to judge how severe these issues are in practice. Therefore, I can not recommend acceptance at this stage.

---

> ### Author Response · Authors · 2021-11-19
> **Response to Reviewer gmFP (Part 1)**
>
> Thanks for reviewing our paper and providing constructive comments. Here are our responses to your concerns.
>
> * **Q1:** *The authors claim that "EVFN is able to represent tensor information losslessly". This is also true for previous equivariant approaches such as TFN, EGNN, PaiNN or NequIP, as long as no distance cutoff is used (see for example Dym and Maron 2020). However, in practice the distance cutoff is essential not only for computational efficiency but also to facilitate generalization across local environments.*
>
> **A1:** No information loss means that we can input all the tensor information into the neural network. To check this, we need to find an inverse map that can recover the original tensor information in the data from our input of the model. `TFN` and `SE(3)-Transformers` achieve that by pairing the input with Spherical Harmonics Function and `EVFN` recovers the tensors by pairing the scalars with the equivariant basis (vectorization). See the discussion at the end of section 3 in the updated version. However, `EGNN` only input the norm of the difference between vectors, from which we cannot recover the position vector field itself. For example, the dihedral angle of the plane spanned by $\vec{x_m} - \vec{x_i}$ and $\vec{x_m} -\vec{x_j}$ and the plane spanned by $\vec{x_m} -\vec{x_i}$ and $\vec{x_m} -\vec{x_k}$ is defined by the inner product of these two planes' normal vector : $\frac{(\vec{x_m} -\vec{x_i}) \times (\vec{x_m} -\vec{x_j})}{||(\vec{x_m} -\vec{x_i}) \times (\vec{x_m} -\vec{x_j})||}$ and $\frac{(\vec{x_m} -\vec{x_i}) \times (\vec{x_m} -\vec{x_k})}{||(\vec{x_m} -\vec{x_i}) \times (\vec{x_m} -\vec{x_k})||}$. Therefore, the dihedral angle cannot be determined by the norms $||\vec{x_m}-\vec{x_i}||$, $||\vec{x_m}-\vec{x_j}||$ and $||\vec{x_m}-\vec{x_k}||$, but rather a function of the three position vectors: $\vec{x_m}-\vec{x_i}$, $\vec{x_m}-\vec{x_j}$ and $\vec{x_m}-\vec{x_k}$ (which is also a function of the scalars in `EVFN`, since `EVFN` can recover the position vectors).
>
> * **Q2:** *The EVFN transform is rotational equivariant and translational invariant only with respect to global transforms.
> Therefore, the proposed representation is not suitable to model general many-particle systems.*
>
> **A2:** As we have clarified in section 2, the equivariance is with respect to global transforms because $\mathbf{X}=(\vec{x_1},\dots,\vec{x_N}) \in \mathbb{R}^{N \times 3}$ is defined as the coordinates of all particles which contains the global information. All the equivariant models cited in this paper assume the global equivariance implicitly. For local equivariance, the transformation group acts at different spaces in different ways (such as gauge symmetry), which is beyond the scope of our paper.
> For the extreme case that two molecules are far from each other, if we have the prior knowledge that the inter-interaction of the two molecules can be neglected, we can apply our model to them separately. If we do not have prior knowledge, then we must consider them as a whole.
>
> * **Q3:** *It would be helpful to apply the proposed approach to the related task of predicting molecular force fields (e.g. on the MD17 dataset where a lot of baselines are available for comparison).*
>
> **A3:** Thank you for your suggestion. Our current experiments including both many-body systems simulations and the benchmark for molecular conformation generation still clearly demonstrate the effectiveness of `EVFN` and these settings are also adopted by many other works in literature. Especially, the datasets GEOM-QM9 and GEOM-DRUGS contain larger and more complex molecular graphs, which are not small systems.
>
> By taking your suggestion, we implemented EVFN on the MD17 dataset following one of the classical baselines, i.e., SchNet [3]. To rigorously validate the expressive power of EVFN, we just leveraged the proposed scalarization block to encode coordinate information into the original SchNet architecture. In the following table, we show that the scalarization block can improve the performance of SchNet on 6 of 8 molecules. Both the SchNet model and the EVFN model are implemented according to the official published codebase (https://github.com/atomistic-machine-learning/schnetpack), trained with the same parameters (i.e., default parameters in the codebase, 6 interaction blocks, 128 feature dimensions and 50k training samples) and evaluated on the same data splitting setting. We are now working on re-running two models on another setting reported in [3] with 3 interaction blocks and 64 feature dimensions. The empirical results will be updated after they are ready. Considering that EVFN could be used to estimate the vector field directly, we are willing to adapt EVFN to directly predict energy-conserving force fields in future work.

---

> > ### Author Response · Authors · 2021-11-19
> > **Response to Reviewer gmFP (Part 2)**
> >
> > **Force MAE for MD17 in meV/Angstrom. All models are trained with 50000 samples.**
> >
> > |Dataset | SchNet | EVFN |
> > | ----| ----| ----|
> > |Aspirin | 11.85 | 12.95 |
> > |Benzene | 10.48 | **7.33**|
> > |Ethanol | 2.73 | **2.57**|
> > |Malonaldehyde | 4.21 | 4.58|
> > |Naphthalene | 9.10 | **7.90**|
> > |Salicylic Acid | 9.99 | **9.50**|
> > |Toluene | 7.11 | **6.84**|
> > |Uracil | 6.31 | **5.90**|
> >
> > * **Q4:** *Questions in the "Further Comments".*
> >
> > **A4:** (1) Thank you for pointing out the interesting periodic boundary setting. To deal with the periodic boundary condition in a form similar with [1] (e.g, an infinite 3D lattice structure), we can move the origin of the coordinate frame to the center of one specific periodic unit (all laws are equally applicable to every periodic unit), and the equivariant basis can be built accordingly.  Then by the nature of periodicity, we can extend the output of our model to other units following the periodic boundary condition.
> >
> > (2) Most of the definitions in the background section have been used in this article. For example, definition 2.1 is used in (3.3) and (3.4) and our scalarization can be seen as a realization of the scalarization technique defined by (2.1).
> >
> > (3) Thank you for pointing this out. We have corrected this citation to Schütt et al (2017) in the current version.
> >
> > (4) First of all, the equivariance of `EVFN` is theoretically guaranteed. We have added a discussion at the end of section 3 to highlight it. We found that the phenomenon is caused by the numerical errors brought by the division operation in the normalization of the basis, which is also mentioned in [2].
> > We fix this issue by increasing the constant $C$ in the denominator, i.e., $\frac{\vec{x}}{||\vec{x}|| + C}$ where ($\vec{x}$ denotes vectors and $C$ denotes the normalization constant to ensure numerical stability). In the following table, we show the impact of different $C$ on the results and increasing $C$ to the same order with `EGNN` can decrease $\Delta_{EQ}$. We have added the updated results in table 1 in the main paper, which is named `EVFN*`.
> >
> > |Setting | Method | $C$ | Inter. | Extra. | Rot. Inter. | Rot. Extra. | $\Delta_{EQ}$|
> > | ---- | ---- | ------ | ---- | ---- | ---- | ---- | ---- |
> > |POS | EGNN | $\mathcal{O}(1)$ | $0.726$ | $6.449$ | $0.726$ | $6.462$ | $9.19 \cdot 10^{-7}$|
> > |         | EVFN | $\mathcal{O}(10^{-3})$ | $0.138$ | $2.502$| $0.137$| $2.428$| $1.23\cdot 10^{-4}$ |
> > |         | EVFN | $\mathcal{O}(1)$ | $0.206$ | $2.432$| $0.206$| $2.442$| $4.13\cdot 10^{-6}$ |
> > |SEFF| EGNN | $\mathcal{O}(1)$ | $0.463$ | $4.248$ | $0.467$ | $4.275$ | $5.88\cdot 10^{-6}$|
> > |         | EVFN |$\mathcal{O}(10^{-3})$| $0.139$ | $1.289$ | $0.142$ | $1.310$ | $1.05\cdot 10^{-6}$|
> > |         | EVFN | $\mathcal{O}(1)$ | $0.278$| $2.406$| $0.289$| $2.556$| $8.17\cdot 10^{-7}$ |
> > |DEFF| EGNN | $\mathcal{O}(1)$ | $1.99\cdot 10^{-2}$ | $0.213$ | $3.22\cdot 10^{-2}$ | $0.290$ | $8.55\cdot 10^{-7}$ |
> > |         | EVFN | $\mathcal{O}(10^{-3})$ | $1.38 \cdot 10^{-3}$ | $2.43 \cdot 10^{-2}$ | $1.39\cdot 10^{-3}$ | $2.48\cdot 10^{-2}$ | $1.15\cdot 10^{-5}$ |
> > || EVFN | $\mathcal{O}(1)$ | $1.76 \cdot 10^{-3}$| $3.02 \cdot 10^{-2}$| $1.76 \cdot 10^{-3}$| $3.04\cdot 10^{-3}$| $6.92\cdot 10^{-7}$ |
> >
> > [1] Schütt K T, Kindermans P J, Sauceda H E, et al. Schnet: A continuous-filter convolutional neural network for modeling quantum interactions[J]. arXiv preprint arXiv:1706.08566, 2017.
> >
> > [2] Garcia Satorras V, Hoogeboom E, Fuchs F, et al. E (n) Equivariant Normalizing Flows[J]. Advances in Neural Information Processing Systems, 2021, 34.

---

> > > ### Comment · Reviewer_gmFP · 2021-11-23
> > > **Thank you for the detailed response**
> > >
> > > The main issue remains Q2/A2: It is clear that the equivariance relates to global transformations, as it is supposed to. However, related approaches use local representations, similar to steerable CNNs, that are able to also capture local, equivariant patterns. This is underlined by the additional results on MD17 as the performance is comparable to what state-of-the-art approaches achieve with 50x less training data. Therefore, I can not change my score.

---

> > > > ### Author Response · Authors · 2021-11-23
> > > > **Response to the further concerns**
> > > >
> > > > Thanks for your response. Could you kindly clarify the definition of "local representations" more clearly?
> > > >
> > > > If "local" refers to the topology relations defined on the molecular graphs or point clouds, EVFN could also capture local interactions with its message passing based Graph-Transformer framework. For the MD17 task, we rigorously followed the implementation details in [1], which takes all molecules as fully-connected graphs due to the small scale of molecules. Therefore, compared to other related works for MD17, we do not understand what extra local information our method would ignore. We hope further clarifications for it.
> > > >
> > > > Considering that this paper is not specifically designed for molecular dynamic scenarios, we didn't incorporate any other priors into our model. However, we do not find any difficulty to combine our equivariant basis with the papers that incorporate much chemical or physical domain knowledge (e.g., DimeNet [2], GemNet [3] and PAINN [4]). We will cite them as related works for many-body system modeling. Moreover, by taking your suggestion, we are exploring and testing our EVFN in the setting with less data. We'll report the results when it is finished and we hope the results can address your concerns.
> > > >
> > > > [1] Schütt K T, Kindermans P J, Sauceda H E, et al. Schnet: A continuous-filter convolutional neural network for modeling quantum interactions[J]. arXiv preprint arXiv:1706.08566, 2017.
> > > >
> > > > [2] Klicpera J, Groß J, Günnemann S. Directional message passing for molecular graphs[J]. arXiv preprint arXiv:2003.03123, 2020.
> > > >
> > > > [3] Klicpera J, Becker F, Günnemann S. GemNet: Universal Directional Graph Neural Networks for Molecules[J]. arXiv preprint arXiv:2106.08903, 2021.
> > > >
> > > > [4] Schütt K T, Unke O T, Gastegger M. Equivariant message passing for the prediction of tensorial properties and molecular spectra[J]. arXiv preprint arXiv:2102.03150, 2021.

---

### Official Review · Reviewer_V2nd · 2021-11-02

**Correctness:** 2
**Technical Novelty And Significance:** 2
**Empirical Novelty And Significance:** Not applicable
**Recommendation:** 5
**Confidence:** 4

**Main Review:**

#### Strengths:
- The paper is well organized and well written (besides some non-standard use of English).
- The mathematical presentation is fairly clear.

#### Weaknesses:
It doesn't seem like the model produces equivariant representations. I agree that the input to the graph transformer model is equivariant to SE(3) transformations, but without some kind of constraint, I don't see how the output could be. Moreover, the empirical evidence does not support this claim (Table 1, col 7).

#### Questions:
- “So far, we have achieved permutation and SE(3) equivariance” (p.5) isn’t it SE(3) invariance? -- the initial centering giving translation invariance, and the scalarization block giving SO(3) invariance
- Did the authors experiment with other kinds of SO(3)-equivariant frames in the scalarization block? I wonder if computing the frame from the neighborhood (e.g., Darboux frames) would improve computational efficiency and/or performance.

#### Possible typos:
- “without concerning about breaking” → without concern about breaking (p. 9)
- “verify the effectiveness of the propose method” → verify the effectiveness of the proposed method (p. 9)
- “The trajectory points are uniformly sampleed” →  The trajectory points are uniformly sampled (p. 6)
- “We sample 1, 30 and 30 trajectories” → We sample 1, ? and 30 trajectories (p. 6)
- “(GNN), which is superior to model” (p. 1)
- “‘Vector filed’” → ‘vector field’ (p.2)
- “We use x(t) to denote” → We use x_i(t) (p.2)
- “Then each point of” → Then each point (p.2)
- “Let {Y^i} be the dual frame of” → … be the dual frame on(?)
- “‘gradient filed’” → gradient field
- “As shown in Table 1, with the original input, EVFN outperforms all other equivariant methods in the interpolation and extrapolation tasks” (p. 7) -- that’s not what table 1 shows.
- “Fij ((a(t), b(t), c(t)))” (p. 4) -- extra parentheses
- Should there be a time index in A5-A8

**Summary Of The Paper:**

The authors introduce a model to predict the time evolution of Newton mechanical systems and small molecules.  The model takes a graphical representation as input and converts it to an SE(3) and permutation equivariant representation using physical principles (i.e., white-box model). This representation is passed through a learned graph transformer module to produce a vector field which is used to predict the time evolution of the system/molecule.

**Summary Of The Review:**

The main claim of the paper is that there is no information lost in the representations learned. As far as I can tell, this claim is not supported by the theory or experiments. Moreover, it does not seem that the learned representations are equivariant.

---

> ### Author Response · Authors · 2021-11-19
> **Response to Reviewer V2nd**
>
> Thanks for reviewing our paper and providing constructive comments. Here are our responses to your concerns.
>
> * **Q1:** *“So far, we have achieved permutation and SE(3) equivariance” (p.5) isn’t it SE(3) invariance? -- the initial centering giving translation invariance, and the scalarization block giving SO(3) invariance.*
>
> **A1:** The statement “So far, we have achieved permutation and SE(3) equivariance” (p.5) is in the vectorization section. After vectorization, the product of the SO(3)-invariant scalar and the equivariant vector yields an SO(3) equivariant vector. We have added a further discussion for a better explanation at the end of section 3 in the updated version.
> To explain the translation equivariance, as we state on page 14: "Note that although the gradient field is supposed to be translation-invariant, the state of the
> many-body system after integrating should be translation-equivariant. Therefore, we add c(0) back for the output of the evolving block."
>
> * **Q2:** *Did the authors experiment with other kinds of SO(3)-equivariant frames in the scalarization block? I wonder if computing the frame from the neighborhood (e.g., Darboux frames) would improve computational efficiency and/or performance.*
>
> **A2:** Thank you for your suggestion. We are not very familiar with the Darboux frame. According to "<https://en.wikipedia.org/wiki/Darboux_frame>", the Darboux frame is associated with a hyper-surface embedded in a high-dimensional space. However, we do not naturally have such a hyper-surface, therefore it seems that it is not easy to transfer Darboux frame directly into our setting.
>
> * **Q3:** *Concerns in the "Weakness": It doesn't seem like the model produces equivariant representations.*
>
> **A3:** **First**, from the theoretical view, we think that we have shown the equivariance of our model from section 3.1 to 3.3 along with appendix A.2, and we have demonstrated that there is no information lost in (3.4): The geometric tensors can be transformed into SO(3)-invariant scalars by the equivariant basis and there is no information lost since we can recover the geometric tensors back from the scalars by pairing the scalars with the equivariant basis. To summarize, the equivariance of our model is based on some obvious principles in group theory,
>
> + the scalarization block transforms the SO(3)-equivariant tensors into SO(3)-invariant scalars;
> + any nonlinear mappings of SO(3) invariant-scalars are still SO(3) invariant-scalars;
> + the vectorization block pairs the SO(3)-invariant scalars with the SO(3)-equivariant basis. From the simple fact that scalars multiply vectors yield SO(3)-equivariant vectors, we can conclude that our model is equivariant.
>
> To make it more clear, we have added a discussion at the end of section 3 for general readers. Also, we have added a formal proof in the appendix of the updated version.
>
> **Second**, we found that the bigger equivariance error of `EVFN` ((Table 1, col 7)) is caused by the numerical errors brought by the division operation in the normalization of the basis, which is also mentioned in [1].
> We fix this issue by increasing the constant $C$ in the denominator, i.e., $\frac{\vec{x}}{||\vec{x}|| + C}$ where ($\vec{x}$ denotes vectors and $C$ denotes the normalization constant to ensure numerical stability). In the following table, we show the impact of different $C$ on the results and increasing $C$ to the same order with `EGNN` can decrease $\Delta_{EQ}$. We have added the updated results in table 1 in the main paper, which is named `EVFN*`.
>
> |Setting | Method | $C$ | Inter. | Extra. | Rot. Inter. | Rot. Extra. | $\Delta_{EQ}$|
> | ---- | ---- | ---- | ---- | ---- | ---- | ---- | ---- |
> |POS | EGNN | $\mathcal{O}(1)$ | $0.726$ | $6.449$ | $0.726$ | $6.462$ | $9.19 \cdot 10^{-7}$|
> |         | EVFN | $\mathcal{O}(10^{-3})$ | $0.138$ | $2.502$| $0.137$| $2.428$| $1.23\cdot 10^{-4}$ |
> |         | EVFN | $\mathcal{O}(1)$ | $0.206$ | $2.432$| $0.206$| $2.442$| $4.13\cdot 10^{-6}$ |
> |SEFF| EGNN | $\mathcal{O}(1)$ | $0.463$ | $4.248$ | $0.467$ | $4.275$ | $5.88\cdot 10^{-6}$|
> |         | EVFN |$\mathcal{O}(10^{-3})$| $0.139$ | $1.289$ | $0.142$ | $1.310$ | $1.05\cdot 10^{-6}$|
> |         | EVFN | $\mathcal{O}(1)$ | $0.278$| $2.406$| $0.289$| $2.556$| $8.17\cdot 10^{-7}$ |
> |DEFF| EGNN | $\mathcal{O}(1)$ | $1.99\cdot 10^{-2}$ | $0.213$ | $3.22\cdot 10^{-2}$ | $0.290$ | $8.55\cdot 10^{-7}$ |
> |         | EVFN | $\mathcal{O}(10^{-3})$ | $1.38 \cdot 10^{-3}$ | $2.43 \cdot 10^{-2}$ | $1.39\cdot 10^{-3}$ | $2.48\cdot 10^{-2}$ | $1.15\cdot 10^{-5}$ |
> || EVFN | $\mathcal{O}(1)$ | $1.76 \cdot 10^{-3}$| $3.02 \cdot 10^{-2}$| $1.76 \cdot 10^{-3}$| $3.04\cdot 10^{-3}$| $6.92\cdot 10^{-7}$ |
>
> We sincerely thank you for pointing out the possible typos, and we have corrected them in the current version.
>
> [1] Satorras V G, Hoogeboom E, Fuchs F B, et al. E(n) Equivariant Normalizing Flows[C]//Thirty-Fifth Conference on Neural Information Processing Systems. 2021.

---

> > ### Comment · Reviewer_V2nd · 2021-11-23
> > **Post rebuttal recommendation**
> >
> > #### Post rebuttal recommendation
> > With consideration of the other reviews and author rebuttal I plan to change my recommendation to weak reject.
> >
> > - The clarification provided by the authors (A3), resolves my concerns about whether the approach is equivariant but the clarity of the paper could be improved.
> >
> > #### Regarding A3
> > - **First** Is it that the coefficients obtained in eq 3.2 are retained and applied in 3.6? If this is the case my concern that the proposed method is not equivariant is resolved; however, this interpretation is not obvious in the current presentation.
> > - **Second** Do you mean you add a term C in the denominator of 3.1?
> >
> > #### Possible typos:
> > - the reference *Victor Garcia Satorras, Emiel Hoogeboom, and Max Welling. E(n) equivariant graph neural networks, 2021b.* appears twice

---

> > > ### Author Response · Authors · 2021-11-23
> > > **Thanks for your response**
> > >
> > > Thank you for raising the rating.
> > >
> > > * For the equivariance of EVFN: we should have indeed clarified this point more clear. And we have added a discussion paragraph in Section 3 and provided more detailed proof in Appendix A.2.3. We will clarify it in the revised main paper.
> > >
> > > * Response to A3: (1) Yes. We applied the coefficients obtained in eq 3.2 into eq 3.6.  No vector update is involved before the pairing in 3.6.
> > > (2) Yes. We implemented this operation following EGNN.
> > >
> > > * Possible typos: Thank you for pointing this out. We will correct it in the revised main paper.

---

### Official Review · Reviewer_YiG5 · 2021-11-03

**Correctness:** 3
**Technical Novelty And Significance:** 2
**Empirical Novelty And Significance:** 3
**Recommendation:** 3
**Confidence:** 5

**Main Review:**

## Strengths
1. Well written and easy to understand.
2. The construction of the equivariant basis seems interesting.

## Weakness
1. The connection between `EVFN` and `EGNN` is not well discussed. Essentially, `EGNN`  takes $||x_i – x_j||^2$ as the geometric scalar and apply $\phi(\cdot)$ to obtain the edgewise message  ( see Eq(3)  in [1]). After obtaining edgewise message, `EGNN`  uses $\phi_x$ to project $m_{ij}$ to a scalar and then vectorizes it by $x_i – x_j$ in Eq(4) in [1]. The key differences of `EVFN` are 1). Extending the equivariant basis from one dimension to three dimensions. 2). Replacing the `MLP` model $\phi_e$ as well as the aggregation operation in Eq(5) and Eq(6) in [1] with an existing `GraphTransformer`  block. In this vein, `EVFN`  can be viewed as the extension of `EGNN` with the new equivariant bases.

2. The motivation behind the equivariant basis construction is not well elaborated.  In the multi-body system modeling, the equivariant basis $\frac{x_i -x_j}{||x_i-x_j||}$ has its physical meaning which indicates the force directions. However,  the benefit of using the outer product of two coordinates to construct the equivariant basis is not clear, since there is no explicit physical meaning of the outer product of two coordinates. It’s better to give more in-depth discussions about the choice of such equivariant bases here.

3. There is no ablation study on the new equivariant basis and the ` GraphTransformer ` on multi-body system modeling.  As mentioned before,  the new-added equivariant bases have no explicit physical meaning of modeling the multi-body system, the main impact factor of the performance improvement of `EVFN` looks vague.
On the other hand, the ablation study on molecular conformation generation even implies that the contribution of ` GraphTransformer ` is slightly larger than that of the new equivariant basis (`EVFN w/o GT` gains the larger performance drop on 5 out of 8 metrics in Table 3. ).  If the performance improvement comes from `GraphTransformer`, the overall contribution of this paper is limited. Therefore, it is crucial to conduct the ablation study of the different components of `EVFN` to show the contributions of the new equivariant basis on multi-body system modeling. For example, apply the new equivariant bases on the `EGNN`  framework.

4. The number of training samples used in the multi-body modeling experiment is not reported. Meanwhile, there is no analysis of the impact of the number of training samples for the different baselines.

5. From Table 1, the result of `GCN`  surpasses many sophisticated baselines in the interpolation and extrapolation tasks, such as `EGNN`.   This is quite weird. Why?


[1] E(n) Equivariant Graph Neural Networks


**Summary Of The Paper:**

This paper proposes a new model: Equivariant Vector Field Network (`EVFN`), which aims to solve the multi-body system modeling problem. Specifically, `EVFN`
 introduces a new kind of equivariant basis which considers more interactions between two particles and employs a graph transformer architecture to learn the edge-wise embeddings.


**Summary Of The Review:**

Overall, I think this paper is not ready to be accepted at this time.

---

> ### Author Response · Authors · 2021-11-19
> **Response to Reviewer YiG5 (Part 1)**
>
> Thanks for reviewing our paper and providing constructive comments. Here are our responses to your concerns.
>
> * **Q1:** *The connection between EVFN and EGNN is not well discussed.*
>
> **A1:** We have discussed the relation with `EGNN` in the second and third paragraph in the introduction, i.e., " `EGNN` directly implements equivariant operations in the original space", "...`EGNN` is abandoning a certain amount of tensor information, ...". In the fourth paragraph, we mention that "`EVFN` first introduces a tuple of complete basis...". We will emphasize the connection more **significantly** in the related works in the updated version by taking your suggestion. Moreover, we'd like to emphasize that extending the equivariant one basis in `EGNN` to the tuple of basis in `EVFN` is non-trivial, because our constructed equivariant basis are **complete** in the sense that the scalarization from these basis covers full information to be used to approximate the ground-truth geometric tensors while `EGNN` does not, e.g., certain angles are functions of the coordinates rather than functions of $||\vec{x_i} - \vec{x_j}||$. For example, the dihedral angle of the plane spanned by $\vec{x_m} - \vec{x_i}$ and $\vec{x_m} - \vec{x_j}$ and the plane spanned by $\vec{x_m} - \vec{x_i}$ and $\vec{x_m} - \vec{x_k}$ is defined by the inner product of these two planes' normal vector : $\frac{(\vec{x_m} - \vec{x_i}) \times (\vec{x_m} - \vec{x_j})}{||(\vec{x_m} - \vec{x_i}) \times (\vec{x_m} - \vec{x_j})||}$ and $\frac{(\vec{x_m} - \vec{x_i}) \times (\vec{x_m} - \vec{x_k})}{||(\vec{x_m} - \vec{x_i}) \times (\vec{x_m} - \vec{x_k})||}$. Therefore, the dihedral angle cannot be determined by the norms $||{\vec{x_m} - \vec{x_i}}||$, $||\vec{x_m} - \vec{x_j}||$ and $||\vec{x_m} - \vec{x_k}||$, but rather a function of the three vectors: $\vec{x_m} - \vec{x_i}$ , $\vec{x_m} - \vec{x_j}$ and $\vec{x_m} - \vec{x_k}$ .
>
>
> * **Q2:** *The motivation behind the equivariant basis construction is not well elaborated... the benefit of using the outer product of two coordinates to construct the equivariant basis is not clear, since there is no explicit physical meaning of the outer product of two coordinates...*
>
> **A2:** Our motivation is mainly from the mathematical point of view - `EGNN` is abandoning a certain amount of tensor information (as shown in the third paragraph in the intro). An example is constructed in A1 for illustration.
> Also, as shown in our experiments, `EGNN` does not perform well in the case that the external force is not along the radial direction, which shows the limited expressiveness of `EGNN`. We choose the cross product to construct the third basis because it is orthogonal to the other two basis and orthogonality is helpful for optimization. Besides, there are many physical laws explicitly contain the cross product, e.g., the electromagnetic force law does contain cross product as an essential ingredient of its expression.
>
> * **Q3:** *There is no ablation study on the new equivariant basis and the GraphTransformer on multi-body system modeling. ... For example, apply the new equivariant bases on the EGNN framework.*
>
> **A3:** We think that our ablation study shown in table 3 is already complete. First, we have applied our new equivariant bases on the `EGNN` framework in table 3, which is named  "`EVFN w/o GT`". The baseline "`EVFN w/o GT`" applies the new equivariant bases on the `GIN` framework used in [1] instead of transformer, and it performs better than "`EVFN w/o Sca`" (which contains transformer but does not contain our extended bases) and vanilla `EGNN` on the GEOM-Drugs dataset.
> Second, the main motivation of "`EVFN w/o Sca`" experiment is to test whether `GraphTransformer` itself can beat `EVFN` by replacing the `GIN` in  [1] with `GraphTransformer`, and the result is obviously negative from table 3.

---

> > ### Author Response · Authors · 2021-11-19
> > **Response to Reviewer YiG5 (Part 2)**
> >
> > * **Q4:** *The number of training samples used in the multi-body modeling experiment is not reported. Meanwhile, there is no analysis of the impact of the number of training samples for the different baselines.*
> >
> > **A4:** (1) We should have indeed made this clearer. As we have described in ''Implementation Details'' of Section 4.1, we sample $1$, $100$ and $100$ trajectories as our evaluation platform, where the data is split into training, validation and test sets by the time span: $T_1 = \lbrace 1.5, 0.5, 0.5 \rbrace$, $T_2 = \lbrace 2, 0.55, 0.55 \rbrace$ and $T_3 = \lbrace 2.5, 0.6, 0.6 \rbrace$. The trajectory points are sampled with the uniform time interval $\Delta t=10^{-3}$ for the POS system and $\Delta t =5 \cdot 10^{-4}$ for the other two systems. Therefore, according to $N_{train} = N_{traj} \cdot \frac{T_1}{\Delta t}$, the number of training labels for three scenarios are $1.5$k, $100$k and $100$k, respectively.
> >
> > (2) Thanks for your good suggestion. Actually, we followed the settings in [2] to conduct the multi-body modeling experiments. By taking your suggestion, we also analyzed the impact of the different number of samples in experiments and added a detailed discussion in appendix A.3.1.
> >
> > * **Q5:** *From Table 1, the result of GCN surpasses many sophisticated baselines in the interpolation and extrapolation tasks, such as EGNN. This is quite weird. Why?*
> >
> > **A5:** **First**, it is not weird that `EGNN` does not perform well, because `EGNN` is superior to model the force along the radial directions and our settings include external forces which are not along the radial directions. **Second**, for the SEFF data without rotation reference, equivariant models do not perform the best because the external force in SEFF is a static constant, which does not rely on coordinates of the nodes while equivariant models model the forces as functions of the coordinates, which increases the difficulties to model a constant external force. However, `GCN` is non-equivariant, so it is unstable under rotated reference (col 5-col 6 in table 1) and data augmentation by rotation.  We have added a footnote in the updated version for the explanation.
> >
> > [1] Shi C, Luo S, Xu M, et al. Learning gradient fields for molecular conformation generation[J]. arXiv preprint arXiv:2105.03902, 2021.
> >
> > [2] Zhuang J, Dvornek N, Li X, et al. Adaptive checkpoint adjoint method for gradient estimation in neural ode[C]//International Conference on Machine Learning. PMLR, 2020: 11639-11649.

---

> > > ### Comment · Reviewer_YiG5 · 2021-11-23
> > > **Thank you for your response**
> > >
> > > Thank you for your response and sorry for the late reply. I've read the response as well as the revised version. Unfortunately, I still decided to hold my score.
> > >
> > > - A1, A2: Thank you for highlighting your contribution that:
> > > > our constructed equivariant basis are complete in the sense that the scalarization from these basis covers full information to be used to approximate the ground-truth geometric tensors while EGNN does not.
> > >
> > > In terms of novelty, I don't think this contribution is enough. In my point of view, **this is the main weakness of this paper**.
> > > Minor:  The cross product in the electromagnetic force law is between the velocity v and magnetic field B, which are not directly relevant to the positions.
> > >
> > > - A3: I'm not satisfied with the A3. I think there is no technical obstacle applying the proposed equivariant basis to `EGNN`.
> > >
> > > - A4: I'm satisfied with this answer.
> > >
> > > - A5: Even though we admit that the external forces may impede the performance of  `EGNN`, it's still wired that the equivariant property does not help to improve the performance since the translation/rotation equivariant are still held even with the external forces.

---

> > > > ### Author Response · Authors · 2021-11-23
> > > > **Response for the further concerns**
> > > >
> > > > * **The novelty of EVFN:**  *In terms of novelty, I don't think this contribution is enough. In my point of view, this is the main weakness of this paper. Minor: The cross product in the electromagnetic force law is between the velocity v and magnetic field B, which are not directly relevant to the positions.*
> > > >
> > > > **Answer1:** (1) By introducing a complete basis, we can transform geometric tensors into scalars in an invertible way (EGNN can only do partial transformation).
> > > > Moreover, `EVFN` can equip any neural network to process these SO(3)-invariant scalars. However, some classical equivariant networks (e.g., `TFN` and `SE(3)-Transformer`) have to adopt restricted architectures to process geometric tensors. `EGNN` can only take rotationally invariant features (e.g., relative distances) as input to avoid breaking SO(3) symmetry.
> > > >
> > > > (2) Actually, for some scenarios that directional information is essential, e.g., molecular dynamics and conformation generation[1], many previous works (e.g., [2] and [3]) are proposed to encode directional information with heavy hand-crafted features (e.g., torsion angles and dihedral angles). Compared to them, `EVFN` proposes a novel way to implicitly encode all directional information, without the demand of hand-crafted features.
> > > >
> > > > **Minor:** We should have indeed made this clear. In practice, we leverage the equivariant bases to encode all type-1 tensors (i.e., vectors) into SO(3)-invariant scalars, which include both coordinates and velocities information. In contrast, EGNN only takes the L2 norm of velocities as the input, ignoring the directional information.
> > > >
> > > > * **Further question for ablation study A3** *I'm not satisfied with the A3. I think there is no technical obstacle applying the proposed equivariant basis to EGNN.*
> > > >
> > > > **Answer2:** **First**, to address your concerns regarding the ablation study, we are willing to apply our equivariant basis into `EGNN` on the molecular generation task and update the results as soon.  However, we are not sure what the exact settings you mean. Therefore, we will build two model variants: `EGNN + Scalarization` and `EGNN + Scalarization + Vectorization`. We think these two experiments could sufficiently demonstrate the effectiveness of the proposed equivariant basis.  If we misunderstand the settings you mean, please further discuss with us.
> > > >
> > > > **Second**, it is reasonable that we didn't include `EGNN` with our equivariant bases in the ablation study, because `EGNN` cannot be naturally taken as the ablative variant of `EVFN`. Our model is built on `ConfGF` [4] that takes `GIN` as the backbone, therefore we also adopt `GIN` as the backbone in the `EVFN w/o GraphTransformer` model.
> > > >
> > > > **Third**, "applying our equivariant bases into `EGNN` (i.e., `EGNN + Scalarization + Vectorization`)" is equivalent to "replacing `GIN` with `GCN` in the `EVFN w/o GraphTransformer` model". We think this slight modification will not change any conclusions in this paper.
> > > >
> > > > * **Further question about A5** *Even though we admit that the external forces may impede the performance of EGNN, it's still wired that the equivariant property does not help to improve the performance since the translation/rotation equivariant are still held even with the external forces.*
> > > >
> > > > **Answer3:** In terms of equivariance, our experiments indeed show the equivariant models generalize better to other frames, which have lower $\Delta_{EQ}$. The multi-body experiment is designed to simulate all possible scenarios with non-radical Newtonian forces in the real world. However, SE(3)-equivariance is not the only prior and other priors such as whether the external force is the function of the equivariant basis (e.g., $\vec{x_i}-\vec{x_j}$) or independent with $\vec{x_i}-\vec{x_j}$ also matter. According to our experiments,  the equivariant models face a trade-off to preserve symmetry. For example, `EGNN` and `EVFN` need to express tensors in the direction of $\vec{x_i}-\vec{x_j}$ or in the coordinate frame determined by the equivariant basis, however, the external force in SEFF is independent with the direction, which increases the difficulty to fit. We will release our code to guarantee reproducibility and further discussions are welcomed.
> > > >
> > > > [1] Klicpera J, Groß J, Günnemann S. Directional message passing for molecular graphs[J]. arXiv preprint arXiv:2003.03123, 2020.
> > > >
> > > > [2] Xu M, Luo S, Bengio Y, et al. Learning neural generative dynamics for molecular conformation generation[J]. arXiv preprint arXiv:2102.10240, 2021.
> > > >
> > > > [3] Klicpera J, Becker F, Günnemann S. GemNet: Universal Directional Graph Neural Networks for Molecules[J]. arXiv preprint arXiv:2106.08903, 2021.
> > > >
> > > > [4] Shi C, Luo S, Xu M, et al. Learning gradient fields for molecular conformation generation[J]. arXiv preprint arXiv:2105.03902, 2021.

---

> > > > > ### Author Response · Authors · 2021-11-30
> > > > > **Further response to ablation study A3**
> > > > >
> > > > > To rigorously validate the expressive power of the proposed equivariant basis, we applied our scalarization module directly on the `EGNN` model, obtaining the model variant `EGNN+Scalarization`. As shown in the following table, for the molecular generation task (QM9), introducing the scalarization module could significantly improve the performance of `EGNN` on 3 of 4 metrics.
> > > > >
> > > > > | Method | COV(Mean)  | COV(Median) | MAT(Mean) | MAT(Median)|
> > > > > | ---- | ---- | ---- | ---- | ---- |
> > > > > |EGNN | **80.93** | 86.27 | 0.3832 | 0.3898|
> > > > > |EGNN+Scalarization | 77.34 | **88.24** | **0.3511**| **0.3388**|

---

### Official Review · Reviewer_BR6H · 2021-11-04

**Correctness:** 2
**Technical Novelty And Significance:** 2
**Empirical Novelty And Significance:** 3
**Recommendation:** 3
**Confidence:** 3

**Main Review:**

Initial Recommendation: Reject

Reason: In my view, in current format, the weaknesses outweigh the strengths of the paper. Please see details below.

Strengths
1. The idea using a scalarization block, a graph transformer and a vectorization block is interesting
2. The results on the synthetic task and the molecular conformer generation task are promising

Weaknesses:
1. For me, the math appears to be imprecise/ wrong at many places which makes it a difficult read - for example, starting in  Def 2.1 shouldn't it  be r vectors and s covectors? Also in the line, the group actions from SO(3) acting should involve the dual representations?
2. For me, the authors make claims without sufficient proofs - for e.g. on page 5 - the bolded line doesn't have a proof - can you please show the same, especially given that the basis associated with ij pair is different. Also the proof for Proposition A.2 is not easy to follow or check.
3. Scalability Concerns - In the tensor example above eq 3.4, the authors provide an example for the s=0 case (2,0) - and this in $\mathbb{R}^3$ alone is of size 9 (3x3) - and this needs to be saved/ computed for each i,j pair. The authors also note that in their case, they only use the (1,0) type tensors. Please add scalability results especially against E(N) GNN which the authors compare against.
4. Appears to be missing some relevant work - for e.g. [1], etc. Moreover the authors only appear to compare their model with the computationally efficient models like E(N) GNN - while ignoring models like LieConv[2] , LieTransformer [3], etc which can be used for the same task (Newtonian Many-Body System task).

References:
1. Anderson, Brandon, Truong-Son Hy, and Risi Kondor. "Cormorant: Covariant molecular neural networks." arXiv preprint arXiv:1906.04015 (2019).
2. Finzi, Marc, et al. "Generalizing convolutional neural networks for equivariance to lie groups on arbitrary continuous data." International Conference on Machine Learning. PMLR, 2020.
3. Hutchinson, Michael J., et al. "Lietransformer: equivariant self-attention for lie groups." International Conference on Machine Learning. PMLR, 2021.

**Summary Of The Paper:**

This work (EVFM) aims to to improve predictions of n-body system dynamics by combining continuous lie symmetries with permutation symmetry. The authors propose to do this by encoding SO(3) invariant representations of each node, followed by the use of a graph transformer and a vectorization block to estimate the vector field. An evolving block is subsequently used to predict dynamics.


**Summary Of The Review:**

Overall, there are certain merits for the proposed architectures, but the paper is hard to read due to some imprecise math, lack of clarity, proofs, etc. I initially suggest rejection.

---

> ### Author Response · Authors · 2021-11-19
> **Response to Reviewer BR6H (Part 1)**
>
> Thanks for reviewing our paper and providing constructive comments. Here are our responses to your concerns.
> * **Q1:** *In Def 2.1, shouldn't it be r vectors and s covectors? The group actions from SO(3) acting should involve the dual representations?*
>
> **A1:** The mathematical notions in our paper follow the standard mathematical monographs and we have given the appropriate references in section 2 (e.g., Def 2.1.10 in [2] for our Def 2.1) for checking. The following are our detailed responses to the questions regarding Def 2.1.
>
> (1) Since covectors may cause ambiguity on representing covariant or contravariant vectors, we take another widely adopted terminology - "vectors" and "dual vectors" for illustration (cf. Def.2.1.10 in [2]), that is "r dual vectors and s vectors". Note that we define the tensor field as a multi-linear map from vectors or dual vectors to a number in $\mathbb{R}^1$, therefore the r vectors in the definition play the role of mapping dual vectors to a number. For example, a (1,0) type tensor $\frac{\partial}{\partial x_i}$ maps a dual vector to a number:
> $\frac{\partial}{\partial x_i}(dx_j) = \delta_{ij}.$
>
> (2) For the "dual representation" you mentioned, do you mean the "dual irreducible representation of a group"? However, we do not apply the irreducible representation of the SO(3) group for building the equivariant network. We only use "vectors" and "dual vectors" to specify how the SO(3) group acts on the tensor fields and we do not mention "the dual representations". To understand how SO(3) acts on the dual vector, please refer to the example in section 1.5 of [3] for an illustration.
>
> If we misunderstand your question, please let us know and we'll try our best to address your further concerns.
>
> * **Q2:** *The bolded line on page 5 doesn't have a proof.*
>
> **A2:** The claim in the bolded line is based on an obvious and simple principle in group theory, that is, if a scalar is SO(3)-equivariant, it must be SO(3)-invariant (cf. the description after eq.(5) in [4], and also in [5] and [6]). Because for $\forall i,j$, the elements in the scalar tuple $t_{ij}$ are scalars, they are invariant under SO(3), i.e., $h(gx) = h(x), \ \ \forall g \in SO(3), h(x)\in t_{ij}$, although the basis associated with ij pair is different.  For the SO(3)-invariant variables, applying any non-linear transformation or a deep neural network $\pi$ to them will not break the symmetry, i.e., $\pi (h(gx)) = \pi (h(x)), \ \ \forall g \in SO(3).$ We have added a discussion section in the end of section 3 for the bolded line in the updated version for general readers.
>
> * **Q3:** *Scalability Concerns: Please add scalability results especially against E(N)-GNN which the authors compare against.*
>
> **A3:** Compared with `EGNN`, the extra cost of `EVFN` is to calculate the values of the scalars in the scalars tuple $t_{ij}, \forall i,j$. Compared to the computational cost of back-propagation in the neural networks, the cost of calculating the $O(3\*3\*N^2)$ (N is the number of nodes in the graph) scalars is much less. For example, for the molecular conformation generation task, transforming the tensors into SO(3)-invariant scalars only brings 9.6\% extra real-time computational cost and 17.4\% extra memory cost.  Although the main purpose of `EVFN` is to construct a complete basis to learn the dynamics with enough expressiveness, we have added the discussion on scalability in A.2.6 in the appendix by taking your suggestions.

---

> > ### Author Response · Authors · 2021-11-19
> > **Response to Reviewer BR6H (Part 2)**
> >
> > * **Q4:** *Appears to be missing some relevant work - for e.g. [1], etc. Moreover the authors only appear to compare their model with the computationally efficient models like E(N) GNN - while ignoring models like LieConv, LieTransformer, etc which can be used for the same task (Newtonian Many-Body System task).*
> >
> > **A4:** We thank the reviewer for providing the reference [1], which is included in the related work section in the updated draft. However, `LieConv` and `LieTransformer` did experiments on the Hamiltonian dynamical system, which outputs a real function (the Hamiltonian) instead of the equivariant vector field. Also, our experiments consider Non-conservative Systems that can not be governed by a Hamiltonian, which are fundamentally different settings from the experiments in `LieConv` and `LieTransformer`.
> >
> > Actually, we have compared with the non-computational efficient equivariant models such as `SE(3)-Transformer`[4]. However, we find that it is hard to jointly optimize the neural ODE and the `SE(3)-Transformer` model. And the empirical results show that `SE(3)-Transformer` achieves an extremely poor performance compared with `EVFN`. Therefore, we did not put the results in our paper.
> >
> > |Setting | Method | Inter. | Extra. | Rot. Inter. | Rot. Extra. | $\Delta_{EQ}$|
> > | ---- | ---- | ---- | ---- | ---- | ---- | ---- |
> > |POS | SE(3)-Transformer | $0.422$ | $4.555$ | $0.442$ | $4.559$ | $7.82 \cdot 10^{-4}$ |
> > |    | EVFN | $0.138$ | $2.502$| $0.137$| $2.428$| $1.23\cdot 10^{-4}$ |
> > |SEFF| SE(3)-Transformer | $12.623$ | $53.358$ | $12.623$ | $53.258$ | $3.67 \cdot 10^{-5}$ |
> >  |   | EVFN | $0.139$ | $1.289$ | $0.142$ | $1.310$ | $1.05\cdot 10^{-6}$ |
> > |DEFF| SE(3)-Transformer | $9.119$ | $37.313$ | $9.119$ | $37.312$ | $3.31 \cdot 10^{-4}$ |
> >  |   | EVFN | $1.38 \cdot 10^{-3}$ | $2.43 \cdot 10^{-2}$ | $1.39\cdot 10^{-3}$ | $2.48\cdot 10^{-2}$ | $1.15\cdot 10^{-5}$ |
> >
> > [1] Anderson, Brandon, Truong-Son Hy, and Risi Kondor. "Cormorant: Covariant molecular neural networks." arXiv preprint arXiv:1906.04015 (2019).
> >
> > [2] Jost J, Jost J. Riemannian geometry and geometric analysis[M]. Berlin: Springer, 2008.
> >
> > [3] Carroll S M. Spacetime and geometry[M]. Cambridge University Press, 2019.
> >
> > [4] Fuchs F B, Worrall D E, Fischer V, et al. Se(3)-transformers: 3d roto-translation equivariant attention networks[J]. arXiv preprint arXiv:2006.10503, 2020.
> >
> > [5] Thomas N, Smidt T, Kearnes S, et al. Tensor field networks: Rotation-and translation-equivariant neural networks for 3d point clouds[J]. arXiv preprint arXiv:1802.08219, 2018.
> >
> > [6] Villar S, Hogg D W, Storey-Fisher K, et al. Scalars are universal: Gauge-equivariant machine learning, structured like classical physics[J]. arXiv preprint arXiv:2106.06610, 2021.

---

> > > ### Comment · Reviewer_BR6H · 2021-11-23
> > > **Reply to authors**
> > >
> > > Thank you very much for the reply and thanks for the clarifications.
> > > However, after reading through the reviews from the other reviewers and corresponding author rebuttals, I would tend to retain my scores.

---

> > > > ### Author Response · Authors · 2021-11-23
> > > > **Further response**
> > > >
> > > > We would appreciate it if you can point out concrete problems or concerns of our paper, including our answers to other reviewers' questions.  Thanks!

---

### Author Response · Authors · 2021-11-19
**Summary**

We appreciate all reviewers for the hard work and helpful comments. We would like to address all reviewers’ concerns in the corresponding responses.

We have updated our manuscript according to the comments. The changes we made mainly include:

* We presented more discussions about the equivariance of our model, the explanation of information lossless in the vectorization subsection, and the discussion of section 3. We gave a formal proof of the equivariance of our model in lemma A.2.3 in the appendix;
* We moved the scalarization paragraph in the background to the appendix and added an illustration for some relevant mathematical symbols above the definition of tensor fields to lighten the burden of the readers;
* We highlighted the relation between `EGNN` and our `EVFN` in related works and a footnote in the introduction.
* We fixed the value of $\Delta_{EQ}$ of `EVFN` (Table 1, col7) and added a detailed discussion in section 4.1 and appendix A.3.1 in the updated draft.
* By taking reviewers' suggestions, we added extra empirical results in appendix A.3.1 to compare with non-computational efficient equivariant models, analyze the impact of the number of training samples and discuss the scalability of `EVFN`.
* Extending the model to local transform (gauge transformation) is added in future work.

We corrected several typos. All changes are marked with red color.

---

### Decision · Program_Chairs · 2022-01-20

**Decision:**

Reject

**Comment:**

The paper proposes a symmetry-informed neural network for modelling many-body systems. The network is empirically evaluated in the tasks of predicting Newtonian trajectories and molecular conformations.

All four reviewers are critical of the paper and recommend rejection (one weak, three strong). The reviews have flagged weaknesses and quality issues with several aspects of the submission, including the proposed methodology, the novelty of the contribution, and the clarity of the presentation. Although detailed clarifications were provided by the authors, most of the reviewers' concerns remain, and the consensus among reviewers remains to reject the paper.

Consequently, the current version of the paper does not appear to meet the quality standards for acceptance to ICLR.